# OCR-Reasoning Benchmark: Unveiling the True Capabilities of MLLMs in Complex Text-Rich Image Reasoning

**Mingxin Huang[†1], Yongxin Shi[†1], Dezhi Peng[*2], Songxuan Lai[2], Zecheng Xie[2], Lianwen Jin[*1]**
[1]South China University of Technology [2]Huawei Technologies Co., Ltd.

## Abstract

Recent advancements in multimodal slow-thinking systems have demonstrated remarkable performance across various visual reasoning tasks. However, their capabilities in text-rich image reasoning tasks remain understudied due to the absence of a dedicated and systematic benchmark. To address this gap, we propose **OCR-Reasoning**, a novel benchmark designed to systematically assess Multimodal Large Language Models on text-rich image reasoning tasks. Specifically, OCR-Reasoning comprises 1,069 human-annotated examples spanning 6 core reasoning abilities and 18 practical reasoning tasks in text-rich visual scenarios. Unlike existing text-rich image understanding benchmarks that only provide a final answer, this benchmark additionally provides a detailed step-by-step reasoning process. This dual annotation enables the evaluation of both the models' final answers and their reasoning processes, thereby offering a holistic assessment of text-rich reasoning capabilities. By leveraging this benchmark, we conducted a comprehensive evaluation of the latest MLLMs. Our results demonstrate that even the most advanced MLLMs exhibit substantial difficulties in text-rich image reasoning tasks, with none achieving an accuracy above 50% on our benchmark, indicating that the challenges of text-rich image reasoning are an urgent issue to be addressed. The benchmark and evaluation scripts are available at https://github.com/SCUT-DLVCLab/OCR-Reasoning.

## 1 Introduction

Recently, slow-thinking systems in Large Language Models (LLMs), such as OpenAI-o1 (Jaech et al., 2024), DeepSeek-R1 (Guo et al., 2025a), Gemini-Thinking (Team et al., 2023), and QwQ (Team, 2025) have demonstrated significant progress in addressing complex math, coding, logical, and scientific problems. Building upon techniques like Chain-of-Thought (CoT) prompting (Wei et al., 2022) and test-time compute scaling (Jaech et al., 2024; Guo et al., 2025a), slow-thinking systems typically engage in critical thinking and reflection before providing the final answer. Moreover, emerging evidence suggests these systems may even experience 'Aha moments' when solving complex problems (Guo et al., 2025a). In order to broaden their ability across diverse contexts, multimodal slow-thinking systems have emerged as a rapidly evolving research direction, driven by the need for more versatile AI applications (Yang et al., 2025; Peng et al., 2025; Meng et al., 2025; Chen et al., 2025b; Liu et al., 2025c; Wang et al., 2025b; Liu et al., 2025b; Shen et al., 2025; Wang et al., 2025a; Liu et al., 2025a).

To assess the reasoning capabilities of multimodal slow-thinking systems, researchers have developed specialized reasoning benchmarks targeting distinct scenarios. For instance, MathVista (Lu et al., 2023), MathVerse (Zhang et al., 2024), Olympiadbench (He et al., 2024), and MathVision (Wang et al., 2024a) are designed to evaluate the math-related reasoning ability of the model. In college-level subject knowledge domains, MMMU (Yue et al., 2023) focuses on advanced reasoning in domains such as chemistry, physics, and scientific problem-solving. While these domains are thriving with the corresponding benchmarks, a critical gap persists in text-rich image scenarios.

---

[†]Equal contribution.
[*]Corresponding authors.

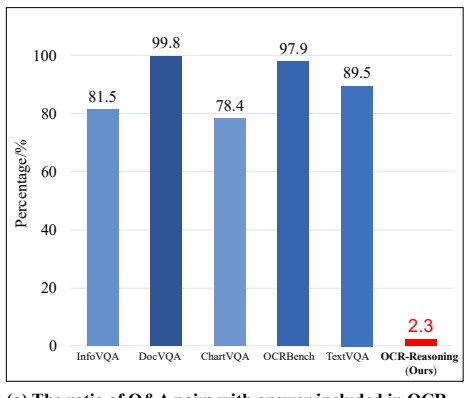
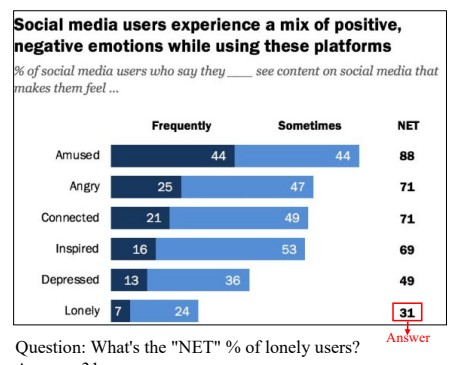

**(a) The ratio of Q&A pairs with answer included in OCR results.**

**(b) Example with answer included in OCR results. The answer can be directly found through fast thinking.**

Figure 1: (a) The percentage of answers in the benchmark's $Q\&A$ pairs that can be retrieved from the OCR results. (b) An example where the answers can be retrieved from the OCR results.

Current benchmarks for text-rich images, such as DocVQA (Mathew et al., 2021), ChartQA (Masry et al., 2022), and OCRBench (Liu et al., 2024d), are designed primarily to assess the ability to merely extract textual content without requiring in-depth analysis (Mathew et al., 2021; Xu et al., 2025). However, text-rich images involve many reasoning-intensive tasks such as financial report analysis, invoice analysis, and cost-effective purchase decisions (Gan et al., 2024; Sun et al., 2021). There is still a lack of benchmarks for systematically evaluating the reasoning ability within text-rich visual scenarios.

To bridge this critical gap in multimodal evaluation, we introduce OCR-Reasoning, a novel benchmark designed to evaluate the text-rich image reasoning skills of Multimodal Large Language Models (MLLMs). Specifically, our benchmark contains 1,069 meticulously collected and human-annotated examples, which span 6 core reasoning abilities and 18 practical reasoning tasks commonly found in text-rich visual contexts. Furthermore, unlike other text-rich image understanding benchmarks that only annotate the final answers, OCR-Reasoning provides annotations for both the final answers and the step-by-step reasoning process. This comprehensive annotation scheme facilitates a more in-depth evaluation of MLLMs' reasoning capabilities. Additionally, through a simple comparison with existing benchmarks, as shown in Fig. 1, we observe that in most cases the answers in existing datasets are directly present in the images, whereas our benchmark contains very few samples of this type. This implies that in our benchmark, to obtain the answer, the model needs to engage in reasoning rather than extracting it from the OCR results of the image.

Using the OCR-Reasoning benchmark, we conduct extensive experiments to assess the text-rich image reasoning capabilities of popular LLMs and MLLMs. For pure LLMs, we replaced images with their OCR results and used these as input. The results show relatively low accuracy, which indicates that text alone is insufficient for solving text-rich image reasoning tasks. For MLLMs, the strongest performer achieves only 46.8% accuracy, with none surpassing 50% on our benchmark. As for document-oriented MLLMs, their highest accuracy does not exceed 15%. These findings demonstrate that existing models still have significant room for improvement in handling text-rich image reasoning tasks. Additionally, we find that most of the existing reinforcement learning methods perform poorly on text-rich image reasoning tasks. Designing reinforcement learning for text-rich image reasoning is a potential direction for enhancing text-rich image reasoning capabilities.

The main contributions of this work are summarized as follows.

- We introduce OCR-Reasoning, a challenging rich-text image reasoning benchmark that provides a systematic evaluation framework for assessing the reasoning capabilities of MLLMs in text-rich scenarios. To the best of our knowledge, we are the first to concretely define various core sub-abilities for text-rich image reasoning and conduct systematic evaluations.

- We conduct a systematic evaluation of leading MLLMs. Our results indicate that: 1) For text-rich image reasoning tasks, pure OCR input cannot effectively replace image input; 2) Even the most leading MLLMs struggle with our proposed benchmark. Based on the experiment results, we find several potential directions for future improvements.

## 2 RELATED WORK

### 2.1 MULTI-MODAL BENCHMARK

Driven by innovations in slow-thinking systems in LLMs, the evaluation of reasoning capabilities in Multimodal Large Language Models (MLLMs) has become a highly focused and widely discussed topic (Lu et al., 2023; Zhang et al., 2024; Wang et al., 2024a; Yue et al., 2023). Early benchmarks such as CLEVR (Johnson et al., 2017) and GQA (Hudson & Manning, 2019) pioneered the integration of compositional language-vision abstraction to assess visual reasoning in structured environments. Subsequent works expanded the evaluation of reasoning into diverse domains. For instance, ScienceQA (Lu et al., 2022) introduces scientific multimodal reasoning requiring domain knowledge. Meanwhile, the emergence of benchmarks like MMMU (Yue et al., 2023) further pushes the boundaries by requiring a college-level reasoning across disciplines like physics and art. With the development of test-time compute scaling (Jaech et al., 2024; Cui et al., 2025), mathematical benchmarks requiring complex reasoning processes to obtain the answer are emerging as critical benchmarks for evaluating the reasoning capabilities of MLLMs. For instance, MathVista (Lu et al., 2023) systematically categorizes seven mathematical reasoning types through multimodal problem decomposition. MathVision (Wang et al., 2024a) curates competition-level mathematical problems with authentic visual contexts. Mathverse (Zhang et al., 2024) introduces a comprehensive multimodal benchmark specifically designed to assess the visual mathematical reasoning capabilities of MLLMs. Although these benchmarks have expanded the scope of evaluation to various domains, there is still a lack of systematic evaluation in the widely applied field of text-rich image understanding. The text-rich image encompasses numerous scenarios requiring reasoning, such as financial report analysis, invoice analysis, cost-effective purchase decisions, and more.

### 2.2 TEXT-RICH IMAGE UNDERSTANDING BENCHMARK

The evolution of Multimodal Large Language Models (MLLMs) has driven corresponding advancements in text-rich image understanding benchmarks. Early benchmarks for text-rich image understanding predominantly focused on assessing the perception capabilities of MLLMs within individual scenarios, such as documents (Mathew et al., 2021), charts (Masry et al., 2022), infographic images (Mathew et al., 2022), and scene text (Singh et al., 2019; Biten et al., 2019). In parallel, recent advancements in high-resolution image processing (Ye et al., 2023; Li et al., 2024c; Huang et al.; Hu et al., 2024b; Guan et al., 2025; Liu et al., 2024a) and optimized computational efficiency (Liu et al., 2024e; Hu et al., 2025; Zhang et al., 2025a; Yu et al., 2024) have significantly improved the performance of these benchmarks. To address the growing need for holistic evaluation of MLLMs, a series of benchmarks with broader, more diverse, and complex scenarios have emerged (Wadhawan et al., 2024; Li et al., 2024a; Liu et al., 2024c;d;b; Ouyang et al., 2024). For instance, OCRBench (Liu et al., 2024d), CC-OCR (Yang et al., 2024b), and OCRBenchv2 (Fu et al., 2024) concentrate on assessing the perceptual capabilities of MLLMs across multiple domains, while OmniDocBench (Ouyang et al., 2024) provides a comprehensive evaluation of PDF document parsing. However, despite these advancements, with the emergence of slow-thinking systems requiring deliberate reasoning, current benchmarks reveal two critical limitations: 1. Overemphasis on textual extraction tasks (Mathew et al., 2021; Xu et al., 2025), which can be solved through fast-thinking processes; 2. Lack of systematic assessment of reasoning capabilities in text-rich image understanding. This progression highlights the pressing need for next-generation benchmarks to evaluate MLLMs' complex reasoning capacities in text-rich visual understanding. To address this limitation, we propose a comprehensive benchmark specifically designed to assess multimodal slow-thinking systems in complex text-rich image reasoning tasks.

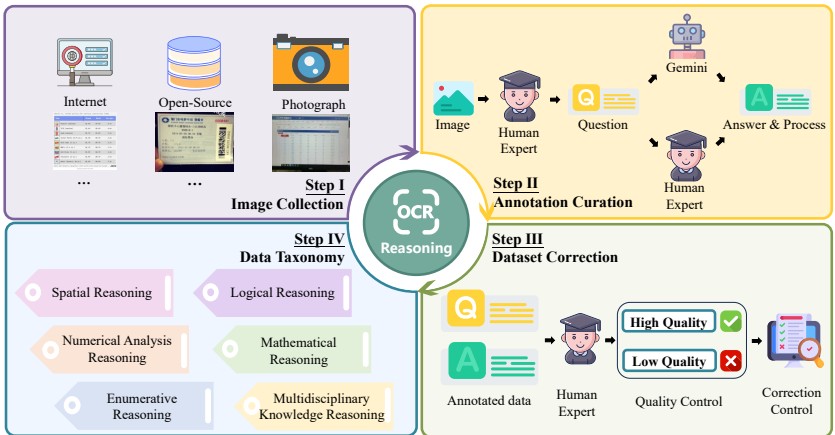

Figure 2: Data curation framework of OCR-Reasoning. The framework includes: (1) dataset collection, (2) annotation curation, (3) data correction, and (4) detailed taxonomy.

# 3  OCR-REASONING

In Sec. 3.1, we first present the data curation framework of OCR-Reasoning, comprising: (1) dataset collection, (2) annotation curation, (3) data correction, and (4) detailed taxonomy. The data curation framework is shown in Fig. 2. Then, in Sec. 3.2, we describe the statistics of OCR-Reasoning, including its total scale, categorical distribution, and detailed question-answer characteristics. Notably, while existing benchmarks (Mathew et al., 2022; Masry et al., 2022; Liu et al., 2024d) focus solely on final answers, OCR-Reasoning provides annotations for both the final answers and the step-by-step reasoning process, facilitating a more in-depth evaluation of MLLMs' reasoning capabilities. The statistics of the annotations are presented in Sec. 3.2.

Additionally, OCR-Reasoning focuses on challenges in single-image. This design choice is based on two well-founded design principles: 1. Capability isolation and focused evaluation: Multi-image or multi-document tasks primarily assess long-context processing capabilities, which require specialized benchmarks for proper evaluation. Mixing single-image reasoning with multi-document challenges would confound the evaluation and make it difficult to isolate specific reasoning deficiencies. 2. Model compatibility and fair evaluation: Several document-oriented MLLMs (Wang et al., 2025c; Xiao et al., 2025; Guan et al., 2025) only focus on single images and have not been trained on multiple images. Including multiple images in the benchmark would exclude these important models from evaluation, potentially reducing the focus on the reasoning problem itself.

## 3.1  DATA CURATION FRAMEWORK

**Dataset Collection.** We constructed the OCR-Reasoning dataset by aggregating images from three primary sources: (1) Internet-sourced images from publicly available online repositories, (2) real-world photographs capturing street views and handwritten notes, and (3) images curated from established benchmarks including InfoVQA (Mathew et al., 2022), DocVQA (Mathew et al., 2021), ChartQA (Masry et al., 2022), CharXiv (Wang et al., 2024c), WildReceipt (Sun et al., 2021), and MME-Finance (Gan et al., 2024). During data collection, we prioritized comprehensive coverage of text-rich scenarios commonly encountered in daily life. We also noted a severe lack of reasoning data related to handwritten content. To address this, our annotators selected and transcribed college-level problems in chemistry, physics, geometry, functions, and statistics, which were then photographed to create a set of handwritten reasoning data. In addition to college-level problems, we also include a portion of handwritten data about logical reasoning tasks. We filtered out those with low resolution or excessive noise. The final dataset comprises 1022 images, a scale comparable to previous reasoning benchmarks (Lu et al., 2023; Bi et al., 2025; Xu et al., 2025). It consists of 476 Internet-sourced images, 253 real-world photographs, and 293 images from established benchmarks.

**Annotation Curation.** After collecting the images, we proceed to annotate them. First, our annotators will design questions based on the images to evaluate the reasoning ability of MLLMs. To

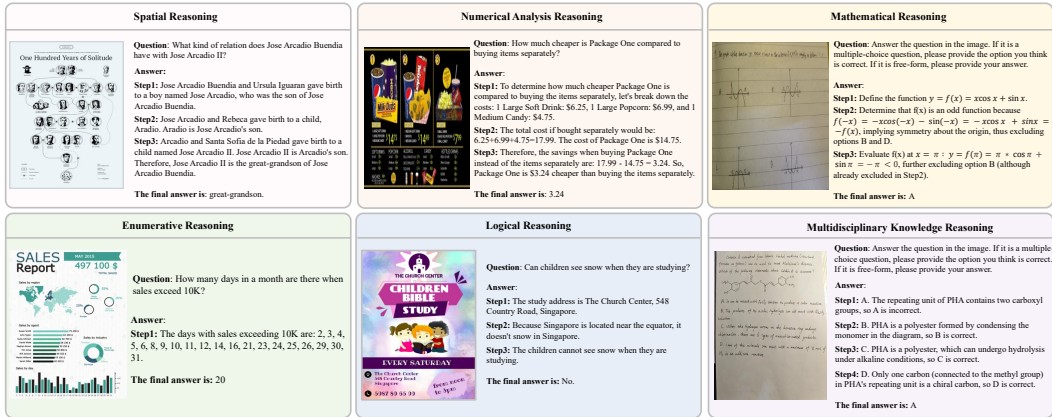

Figure 3: Examples of different categories in OCR-Reasoning. OCR-Reasoning includes six categories: spatial Reasoning, numerical analysis reasoning, mathematical reasoning, enumerative reasoning, logical reasoning, and multidisciplinary knowledge reasoning.

Figure 4: Subject Distribution of OCR-Reasoning.

Table 1: Key Statistics of OCR-Reasoning.

| Statistic | Number |
|---|---|
| Total questions | 1069 |
| - Multiple-choice questions | 250 (23.4%) |
| - Free-form questions | 819 (76.6%) |
| - Newly collected question | 987 (92.3%) |
| - Newly collected reasoning path | 1069 (100.0%) |
| Number of unique images | 1022 |
| Number of unique questions | 1069 |
| Number of unique answers | 1069 |
| Maximum question length | 393 |
| Maximum answer length | 3106 |
| Average question length | 76 |
| Average answer length | 421 |

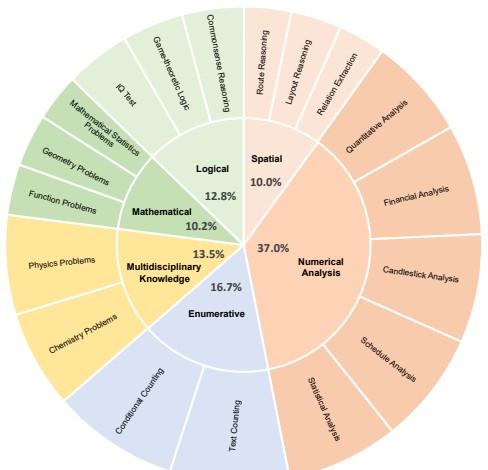

guarantee the quality of the data, we engage PhD candidates in STEM fields as expert annotators. For each image, three annotators independently propose a question. Then, other annotators score and select the highest-quality question. Subsequently, we generate reasoning processes and answers through two parallel pathways: 1. Human Annotation: Annotators manually produce one reasoning process along with the corresponding answer. 2. Model-Based Generation: We input both the questions and answers into closed-source MLLMs (e.g., Gemini 2.0 Flash) to generate an alternative reasoning process and answer.

**Data Correction.** After obtaining the questions, reasoning processes, and answers, three annotators evaluate the annotations from both pathways. The pathway with the highest average score is chosen as the final reasoning process and the corresponding answers. Finally, a manual review step is conducted to examine and correct all question-answer pairs and reasoning processes.

**Data Taxonomy.** After completing the data annotation process, we will categorize the data into six categories based on the reasoning skills required to answer the questions. To mitigate human bias, we implement a two-stage classification approach. In the initial phase, three annotators independently classified each example into one of six predefined categories. Then, we implemented a majority voting system where the final category assignment was determined by plurality consensus among the three annotators. The definitions of each category are as follows: **Spatial Reasoning** focuses on the model's ability to reason about spatial relationships between text and visual elements, as

well as layout elements within text-rich images. **Numerical Analysis Reasoning** involves calculations related to numerical variations in text-rich images, including cost-effective purchase decisions, growth rate estimation, financial report analysis, schedule planning, and data interpretation. Numerical Analysis Reasoning also encompasses samples related to scenarios involving web screenshots, financial documents, or product manuals. **Mathematical Reasoning** entails solving mathematical problems (e.g., functions, geometry, statistics) in text-rich images using mathematical knowledge. Compared to other mathematical benchmarks, the mathematical reasoning-related data in our benchmark is handwritten by our annotators, which requires models to possess stronger OCR capabilities to accomplish these tasks. **Enumerative Reasoning** focuses on counting text and visual elements in text-rich images that satisfy specific query conditions. **Logical Reasoning** requires critical thinking and drawing inferences from provided text-rich images to arrive at conclusions. **Multidisciplinary Knowledge Reasoning** involves applying cross-domain knowledge (e.g., physics, chemistry) to interpret text and visual elements in text-rich images. We provide some examples in Fig. 3.

## 3.2 DATASET STATISTICS

The key statistics of OCR-Reasoning are summarized in Tab. 1. This benchmark contains 1,069 questions categorized into two distinct formats: multiple-choice (with provided answer options) and free-form responses. The free-form answers are further classified into three data types: integers, floating-point numbers, and strings. Notably, our benchmark contain extended analytical reasoning processes, evidenced by an average combined length of 421 characters for reasoning chains and final answers. The maximum length reaches 3,106 characters, highlighting the complexity of the OCR-Reasoning. As shown in Fig. 4, the question distribution spans six reasoning categories: Spatial Reasoning (10.0%), Numerical Analysis (37.0%), Logical Reasoning (12.8%), Mathematical Reasoning (10.2%), Multidisciplinary Knowledge (13.5%), and Enumerative Reasoning (16.7%). Numerical Analysis Reasoning covers 5 real-world task types, more than the 2-3 task types in other categories, hence it accounts for a larger proportion. More examples are presented in the Sec. A.8.

## 3.3 EVALUATION PROTOCOLS

Following previous methods (Lu et al., 2023; Zhang et al., 2024), OCR-Reasoning adopts a three-stage evaluation framework: (1) Response Generation, (2) Answer Extraction, and (3) Score Computation. First, the multimodal large language model (MLLM) processes an input query to generate detailed responses. Subsequently, an LLM-based answer extractor (e.g., GPT-4o) extracts concise answer text from these responses through semantic parsing. Our preliminary study on 200 examples shows that this extraction process achieves over 99.5% accuracy. Finally, the extracted answers undergo normalization into standardized formats (e.g., option letters, integers, or strings) before accuracy-based metric calculation for deterministic evaluation.

For the evaluation of reasoning processes, inspired by evaluation in large language models (Zheng et al., 2023; Chang et al., 2024), we employed the LLM-as-judge (Zheng et al., 2023) approach to assess the reasoning process. Given a question, a detailed response from an MLLM, and a ground truth of the reasoning trajectory, an LLM judge is asked to directly assign a score to detailed responses. Our adoption of this methodology is based on solid empirical justification: 1) Human evaluation is costly, while LLM can quickly process large amounts of data. 2) LLM can reduce the variance among human evaluators. 3) LLM as Judge is commonly used in NLP to evaluate reasoning processes. We tried to use human-grounded validation across different models to compare their scores with those of the LLM-as-Judge: the human-grounded validation score for DouBao-1.5-Vision-Pro is 53.1 (vs. LLM-as-Judge score 55.4), for Qwen2.5-VL-72B is 50.2 (vs. LLM-as-Judge score 51.8), for Llama4-Scout-109B-A17B is 43.8 (vs. LLM-as-Judge score 44.9), and for OpenAI-o1 is 47.6 (vs. LLM-as-Judge score 48.5). The scores of human-grounded validation are close to those of the LLM-as-Judge.

## 4 EXPERIMENT

In this section, we conduct a comprehensive evaluation of existing MLLMs on OCR-Reasoning. We first describe the experimental setup in Sec. 4.1. Then, the overall results and the corresponding analysis are presented in Sec. 4.2.

Table 2: Accuracy scores on the OCR-Reasoning. The results include OCR + LLM, closed-source MLLMs, and open-source MLLMs. Bold denotes the best performance.

| Method | Overall | Spatial | Numerical Analysis | Mathematical | Enumerative | Logical | Multidisciplinary Knowledge |
|---|---|---|---|---|---|---|---|
| **OCR + LLM** | | | | | | | |
| OpenAI-o3-mini (Open AI Team, 2025) | **33.3** | **17.4** | **41.2** | **25.5** | **41.3** | 24.3 | 27.7 |
| DeepSeek-R1-Distill-Qwen-32B (Guo et al., 2025a) | 26.5 | 11.9 | 28.9 | 23.5 | 34.6 | 18.8 | 30.7 |
| Qwen2.5-32B Yang et al. (2024a) | 26.5 | 13.7 | 29.9 | 16.6 | 29.1 | **26.4** | **31.4** |
| **Closed-Source MLLM** | | | | | | | |
| Gemini-2.0-Flash (Team et al., 2023) | 39.3 | 19.3 | 47.2 | 24.5 | 49.7 | 36.8 | 32.1 |
| GPT-4o (Hurst et al., 2024) | 30.7 | 21.1 | 35.9 | 18.6 | 40.8 | 26.4 | 23.4 |
| OpenAI-o1 (Jaech et al., 2024) | 44.4 | **27.5** | 46.2 | **43.1** | 50.8 | **40.3** | 49.6 |
| Claude-3.7-Sonnet (Anthropic, 2025) | 35.8 | 20.2 | 35.4 | 23.5 | **60.3** | 30.6 | 32.1 |
| DouBao-1.5-Vision-Pro (Guo et al., 2025b) | **46.8** | 27.5 | **54.0** | 33.3 | 50.8 | 34.7 | **58.4** |
| **Open-Source MLLM** | | | | | | | |
| Qwen2.5-VL-3B (Bai et al., 2025) | 12.2 | 11.0 | 11.8 | 9.8 | 19.0 | 7.6 | 11.7 |
| Qwen2.5-VL-7B (Bai et al., 2025) | 15.7 | 13.8 | 11.6 | 8.8 | 20.1 | 9.0 | 35.8 |
| Qwen2.5-VL-32B (Bai et al., 2025) | 36.2 | 21.1 | 38.7 | 25.5 | 46.9 | 34.7 | 36.5 |
| Qwen2.5-VL-72B (Bai et al., 2025) | 37.5 | **24.8** | 44.7 | 22.5 | 47.5 | 28.5 | 34.3 |
| InternVL3-2B (Zhu et al., 2025) | 10.8 | 11.9 | 4.8 | 7.8 | 18.4 | 11.8 | 18.3 |
| InternVL3-8B (Zhu et al., 2025) | 11.5 | 12.8 | 5.8 | 11.8 | 17.9 | 7.6 | 22.6 |
| InternVL3-32B (Zhu et al., 2025) | 17.1 | 14.7 | 10.3 | 14.7 | 24.0 | 11.8 | 37.2 |
| InternVL3-78B (Zhu et al., 2025) | 19.9 | 13.8 | 22.4 | 9.8 | 14.0 | 27.1 | 25.5 |
| Llama4-Scout-109B-A17B (Meta, 2025) | 27.7 | 15.6 | 34.7 | 16.7 | 41.3 | 22.9 | 12.4 |
| Kimi-VL-A3B-Thinking (Team et al., 2025a) | 20.5 | 11.9 | 22.4 | 14.7 | 24.6 | 21.5 | 19.7 |
| VL-Rethinker-7B (Wang et al., 2025a) | 14.6 | 8.3 | 16.1 | 9.8 | 19.6 | 8.3 | 19.0 |
| MM-Eureka-Qwen-7B (Meng et al., 2025) | 13.2 | 9.2 | 7.0 | 10.8 | 18.4 | 15.3 | 27.0 |
| VLAA-Thinker-Qwen2.5VL-7B (Chen et al., 2025a) | 14.4 | 11.9 | 10.3 | 7.8 | 21.2 | 11.8 | 27.0 |
| QvQ (Wang et al., 2024b) | 32.7 | 24.8 | 34.7 | 15.7 | 44.1 | 31.9 | 32.1 |
| Keye-VL-8B (Team et al., 2025b) | 22.6 | 13.8 | 21.9 | 26.5 | 25.7 | 17.4 | 30.7 |
| Thyme-RL-7B (Zhang et al., 2025b) | 15.2 | 12.8 | 10.8 | 10.8 | 20.7 | 18.1 | 23.4 |
| DeepEyesV2 (Hong et al., 2025a) | 20.9 | 11.9 | 18.8 | 13.7 | 27.9 | 18.85 | 32.8 |
| MiMo-VL-RL-7B (Xiaomi, 2025) | 38.8 | 20.2 | 41.2 | 22.5 | 51.4 | **38.9** | 42.3 |
| GLM-4.1V-Thinking-9B (Hong et al., 2025b) | **44.1** | 22.9 | **49.2** | **35.3** | **53.1** | 35.4 | **50.4** |
| **Document-Oriented MLLMs** | | | | | | | |
| mPLUG-DocOwl2-8B (Hu et al., 2024a) | 3.3 | 3.7 | 0.3 | 1.0 | 7.3 | 9.7 | 1.5 |
| Docopilot-8B (Duan et al., 2025) | 11.6 | 11.9 | 6.5 | 6.9 | 19.6 | 8.3 | 22.6 |
| DocMark-2B (Xiao et al., 2025) | 7.4 | 8.3 | 0.3 | 6.9 | 14.5 | 3.5 | 22.6 |
| TokenVL-8B (Guan et al., 2025) | 14.3 | 10.1 | 8.8 | 8.9 | 25.7 | 13.9 | 23.4 |

## 4.1 EXPERIMENT SETUP

**Evaluation Models.** We evaluate three distinct types of foundation models on OCR-Reasoning: (a) Large Language Models (LLMs) with OCR results (Extracting by PP-OCRv3 (Li et al., 2022)), including Deepseek-R1 (Zhou et al., 2025) and OpenAI-o3-mini. (b) closed-source MLLMs, comprising Gemini-2.0-Flash (DeepMind, 2025), GPT-4o (Hurst et al., 2024), OpenAI-o1 (Jaech et al., 2024), Claude-3.7-Sonnet (Anthropic, 2025), and DouBao-1.5-Vision-Pro (Guo et al., 2025b). (c) Open-source MLLMs, represented by models like Qwen2.5-VL (Bai et al., 2025), InternVL3 (Zhu et al., 2025), Llama4-Scout (Meta, 2025), Kimi-VL (Team et al., 2025a), VL-Rethinker (Wang et al., 2025a), MM-Eureka (Meng et al., 2025), VLAA-Thinker (Chen et al., 2025a). (d) Document-Oriented MLLMs, including mPLUG-Docow2 (Hu et al., 2025), Docopilot (Duan et al., 2025), DocMark (Xiao et al., 2025), and TokenVL (Guan et al., 2025).

**Implementation Details.** To evaluate the generalization capacity of Multimodal Large Language Models (MLLMs), we adopt a zero-shot evaluation protocol without model fine-tuning or few-shot prompting. Following the standardized chain-of-thought paradigm, we present MLLMs with both visual inputs (images) and textual questions, accompanied by explicit instructions: "Solve the complex problem through step-by-step reasoning." For text-only Large Language Models (LLMs), we substitute visual inputs with the OCR results (using PP-OCRv3 (Li et al., 2022) to obtain the OCR results) while retaining identical textual queries. Given the inherent variability in output formats across text-rich image scenarios (e.g., monetary values like $15, temporal expressions like 20 days, or timestamps like 19:00:00), we implement format-specific prompting. This involves appending the directive: "The composition of the final answer should be: xxxxx" to each query. For instance, when expecting currency outputs "$15", the format-specific prompting is: "The composition of the final answer should be: $ + Integer".

Table 3: Impact of Chain-of-Thought prompting on different MLLMs.

| Method | CoT | Overall | Spatial | Numerical Analysis | Mathematical | Enumerative | Logical | Multidisciplinary Knowledge |
|---|---|---|---|---|---|---|---|---|
| Qwen2.5-VL-32B (Bai et al., 2025) | × | 33.0 | 12.8 | 33.7 | 24.5 | 48.0 | 28.4 | 38.7 |
| Qwen2.5-VL-32B (Bai et al., 2025) | ✓ | 36.2 | 21.1 | 38.7 | 25.5 | 46.9 | 34.7 | 36.5 |
| GPT-4o (Hurst et al., 2024) | × | 26.5 | 11.9 | 33.4 | 15.7 | 29.1 | 25.0 | 24.1 |
| GPT-4o (Hurst et al., 2024) | ✓ | 30.7 | 21.1 | 35.9 | 18.6 | 40.8 | 26.4 | 23.4 |
| Kimi-VL-A3B-Thinking (Team et al., 2025a) | × | 20.1 | 11.0 | 19.1 | 16.7 | 30.2 | 19.4 | 20.4 |
| Kimi-VL-A3B-Thinking (Team et al., 2025a) | ✓ | 20.5 | 11.9 | 22.4 | 14.7 | 24.6 | 21.5 | 19.7 |
| VL-Rethinker-7B (Wang et al., 2025a) | × | 19.1 | 13.7 | 16.6 | 9.8 | 25.7 | 14.6 | 33.6 |
| VL-Rethinker-7B (Wang et al., 2025a) | ✓ | 14.6 | 8.3 | 16.1 | 9.8 | 19.6 | 8.3 | 19.0 |
| MM-Eureka-Qwen-7B (Meng et al., 2025) | × | 12.2 | 10.1 | 6.3 | 8.8 | 16.8 | 14.6 | 25.5 |
| MM-Eureka-Qwen-7B (Meng et al., 2025) | ✓ | 13.2 | 9.2 | 7.0 | 10.8 | 18.4 | 15.3 | 27.0 |

Table 4: Reasoning scores on the OCR-Reasoning benchmark. Bold denotes the best performance.

| Method | Overall | Spatial | Numerical Analysis | Mathematical | Enumerative | Logical | Multidisciplinary Knowledge |
|---|---|---|---|---|---|---|---|
| **Closed-Source MLLM** | | | | | | | |
| Gemini-2.0-Flash (Team et al., 2023) | 49.5 | 31.5 | 57.1 | 42.6 | 49.3 | 47.4 | 49.2 |
| GPT-4o (Hurst et al., 2024) | 45.4 | 35.4 | 48.9 | 33.0 | 48.7 | 48.0 | 45.5 |
| OpenAI-o1 (Jaech et al., 2024) | 48.5 | 36.9 | 53.9 | 50.0 | 39.4 | 49.4 | 51.8 |
| Claude-3.7-Sonnet (Anthropic, 2025) | 50.3 | 37.7 | 55.0 | 38.8 | **58.1** | 48.6 | 46.5 |
| DouBao-1.5-Vision-Pro (Guo et al., 2025b) | **55.4** | 38.2 | **61.8** | 50.2 | 52.4 | 52.8 | **61.2** |
| **Open-Source MLLM** | | | | | | | |
| Qwen2.5-VL-3B (Bai et al., 2025) | 22.3 | 18.5 | 25.6 | 15.7 | 22.9 | 20.8 | 21.3 |
| Qwen2.5-VL-7B (Bai et al., 2025) | 34.0 | 24.9 | 39.2 | 27.5 | 41.5 | 28.9 | 27.3 |
| Qwen2.5-VL-32B (Bai et al., 2025) | 54.6 | **38.5** | 58.9 | 45.9 | 55.8 | **56.3** | 57.9 |
| Qwen2.5-VL-72B (Bai et al., 2025) | 51.8 | 35.9 | 58.6 | 41.3 | 54.7 | 49.9 | 50.8 |
| InternVL3-2B (Zhu et al., 2025) | 15.7 | 15.3 | 13.4 | 13.3 | 21.4 | 16.5 | 15.7 |
| InternVL3-8B (Zhu et al., 2025) | 16.3 | 16.1 | 14.9 | 13.8 | 21.5 | 15.0 | 16.6 |
| InternVL3-32B (Zhu et al., 2025) | 42.6 | 32.1 | 43.9 | 38.0 | 45.3 | 43.6 | 46.1 |
| InternVL3-78B (Zhu et al., 2025) | 43.3 | 29.2 | 50.3 | 35.5 | 38.6 | 46.9 | 42.2 |
| Llama4-Scout-109B-A17B (Meta, 2025) | 44.9 | 33.0 | 49.3 | 36.7 | 47.1 | 45.4 | 44.1 |
| Kimi-VL-A3B-Thinking (Team et al., 2025a) | 40.8 | 30.9 | 43.1 | 37.4 | 45.6 | 38.5 | 40.7 |
| VL-Rethinker-7B (Wang et al., 2025a) | 29.8 | 23.1 | 33.7 | 23.7 | 31.4 | 26.5 | 29.8 |
| MM-Eureka-Qwen-7B (Meng et al., 2025) | 21.9 | 20.7 | 20.5 | 16.5 | 24.8 | 22.2 | 26.9 |
| VLAA-Thinker-Qwen2.5VL-7B (Chen et al., 2025a) | 24.8 | 22.2 | 25.8 | 18.0 | 25.9 | 24.2 | 28.5 |
| **Document-Oriented MLLMs** | | | | | | | |
| mPLUG-DocOwl2-8B (Hu et al., 2024a) | 12.9 | 13.6 | 12.6 | 10.3 | 13.8 | 15.6 | 10.9 |
| Docopilot-8B (Duan et al., 2025) | 20.6 | 17.1 | 22.4 | 13.2 | 25.2 | 19.6 | 19.1 |
| DocMark-2B (Xiao et al., 2025) | 15.3 | 13.9 | 13.5 | 12.3 | 22.7 | 15.3 | 13.9 |

## 4.2 OVERALL RESULTS

**The use of visual images as input is crucial.** To assess its importance, we replace the images with their OCR results and feed them into the LLMs for comparison. As shown in Tab. 2, substituting image input with OCR text leads to a significant decline in model performance. For instance, when using the same LLM, the performance of even a strong reasoning model like DeepSeek-R1-Distill-Qwen-32B remains 9.7% lower than that of Qwen2.5-VL-32B. This demonstrates the critical importance of image input for text-rich image reasoning tasks. We present some qualitative results in Appendix A.3.

**The performance of current MLLMs still has significant room for improvement.** Specifically, our analysis reveals several key observations: 1. As shown in Tab. 2, the top-performing model is Doubao-1.5-Vision-Pro. While Doubao-1.5-Vision-Pro performs strongly on text-rich image understanding tasks—such as DocVQA (96.7%), InfoVQA (89.3%), and ChartQA (87.4%)—its performance on OCR-Reasoning does not exceed 50%. This highlights the particular challenge of integrating visual, textual, and logical information in reasoning scenarios. 2. Among different reasoning types, MLLMs perform most strongly on enumerative reasoning, which consistently ranks as the first or second best capability in both closed-source and open-source models. 3. Furthermore, scaling up model parameters is positively correlated with performance gains, as illustrated by the Qwen2.5-VL series: the 7B model surpasses the 3B version by 3.5%, and the 32B model outperforms the 7B version by 20.5%. 4. Additionally, we observe that document-oriented MLLMs still face difficulties in complex reasoning. Although document-oriented MLLMs are effective at basic comprehension, their limitations in deeper reasoning underscore the need for innovations in model architecture or training strategies.

**CoT prompting performs differently across different models.** The results of the influence of CoT prompts are presented in Tab. 3. We observe that CoT prompting performs differently across

Table 5: Impact of few-shot prompting on Qwen2.5-VL-7B.

| Method | Overall | Spatial | Numerical Analysis | Mathematical | Enumerative | Logical | Multidisciplinary Knowledge |
|---|---|---|---|---|---|---|---|
| Qwen2.5-VL-7B (Bai et al., 2025) | 15.7 | 13.8 | 11.6 | 8.8 | 20.1 | 9.0 | 35.8 |
| One-shot prompting | 16.1 | 12.8 | 14.6 | 10.8 | 22.3 | 13.9 | 21.1 |
| Three-shot prompting | 16.4 | 13.7 | 14.8 | 10.0 | 22.9 | 13.2 | 22.3 |

different models. On most models, CoT prompting consistently enhances their capabilities. This improvement is particularly pronounced in spatial reasoning, where CoT prompting exerts the most significant impact. Specifically, CoT prompting improves performance by 3.2% on Qwen2.5-VL-32B, and 4.2% on GPT-4o, respectively. However, for VL-Rethinker-7B (Wang et al., 2025a), the application of CoT prompting typically results in performance degradation. This phenomenon may stem from the forced rethinking machine on VL-Rethinker-7B. Adding an additional CoT prompt during inference creates a discrepancy between training and testing conditions, ultimately leading to reduced performance.

**Impact of Reasoning path.** The scores of the reasoning path are presented in Tab. 4. Overall, we observe that the ranking based on Reasoning Path scores aligns with that based on final answer accuracy, with the exception of Gemini and Claude-3.7-Sonnet. Specifically, the high scores of Gemini and Claude-3.7-Sonnet are primarily due to the high quality of their reasoning path. We find that for many erroneous samples, the reasoning steps are largely sound, with only minor errors leading to the incorrect outcome. Additional qualitative analyses are provided in the appendix (Sec. A.2) to further illustrate these observations.

**Some reinforcement learning methods perform poorly on text-rich image reasoning tasks.** The performance of some reinforcement learning methods on text-rich image reasoning tasks is relatively poor compared to their baseline. There are several possible reasons for this. First, the reward function: The reward functions in these reinforcement learning methods are not specifically designed for text-rich image reasoning tasks. Most existing reward functions are tailored for mathematical reasoning tasks. How to design a reward function applicable to text-rich image reasoning tasks is a highly worthwhile research direction. Second, a notable discrepancy exists between the training data and the benchmark. The majority of training data is primarily designed for printed mathematical problems, while our benchmark contains data from a wide variety of scenarios. How to select training data to improve OCR inference performance is a highly valuable research direction.

**Impact of Few-shot Prompting.** We also conducted experiments to explore the impact of few-shot prompting. We annotated three additional samples as few-shot demonstrations and validated the performance of one-shot and three-shot prompting on Qwen2.5-VL-7B. As shown in the Tab. 5, few-shot prompting improves overall performance—particularly on subtasks requiring adherence to specific logical steps (e.g., Numerical Analysis Reasoning and Logical Reasoning). This demonstrates that moderate task-specific guidance helps the model understand and comply with task requirements. We observed a decline in the performance of Multidisciplinary Knowledge Reasoning. The potential reasons may be: The increased length of input tokens caused by few-shot examples, combined with the extended reasoning path inherently required by multidisciplinary knowledge reasoning, poses a significant challenge to the model's long-text processing and reasoning capabilities.

**The "thinking with images" approach represents a promising direction for enhancing reasoning capabilities over text-rich images.** "Thinking with images" has demonstrated tremendous potential in general-scene reasoning; therefore, we also evaluated the two latest "thinking with images" methods on the OCR-Reasoning task. The experimental results are presented in Table 2, which illustrate the potential of this approach to enhance the model's reasoning ability on text-rich images.

## 4.3 COMPARISON WITH EXISTING BENCHMARKS

We conducted experiments to quantitatively compare our OCR-Reasoning dataset with existing benchmarks (including DocVQA, ChartQA, TextVQA, OCRBench) using three multimodal large language models (MLLMs): Qwen2.5-VL-7B, InternVL3-8B, and TokenVL-8B. The experimental results are presented in Tab. 6. As observed, while existing methods achieve strong performance

on conventional datasets, they exhibit significant performance degradation on text-rich image reasoning tasks. This discrepancy stems from the fundamental difference in evaluation focus: existing datasets primarily assess models' perceptual capabilities (e.g., text detection, recognition, and basic visual-linguistic alignment), whereas OCR-Reasoning requires the model to achieve accurate perception and further conduct thinking and reasoning. These findings indicate that the text-rich image reasoning ability of current MLLMs still has substantial room for improvement.

Table 6: Comparison with Existing Benchmarks

| Models | DocVQA | ChartQA | OCRBench | TextVQA | OCR-Reasoning |
|---|---|---|---|---|---|
| Qwen2.5-VL-7B | 95.7 | 87.3 | 864 | 84.9 | 15.7 |
| InternVL3-8B | 92.7 | 86.6 | 880 | 80.2 | 11.5 |
| TokenVL-8B | 94.2 | 86.6 | 860 | 79.9 | 14.3 |

## 4.4 ERROR ANALYSIS

We analyzed the error types of DouBao-1.5-Vision-Pro and classified them into six major categories: (1) calculation errors (37.5%), (2) spatial comprehension errors (27.2%), (3) logical errors (19.9%), (4) text perception errors (7.7%), (5) knowledge application errors (5.6%), and (6) summary errors (3.1%). Calculation errors occur when layout understanding and text perception are correct, but mistakes are made during the calculation process. Spatial comprehension errors arise when the model fails to properly understand the spatial layout information of text in images. Logical errors result from unreasonable assumptions or inverted cause-and-effect relationships. Knowledge application errors occur in cases such as the incorrect application of theorems or misunderstanding of common sense. Text perception errors happen when text recognition is incorrect. Summary errors occur when the reasoning process is normal but the final answer is incorrect. This distribution reveals that the DouBao-1.5-Vision-Pro's primary challenges lie in higher-level cognitive tasks rather than basic perception tasks.

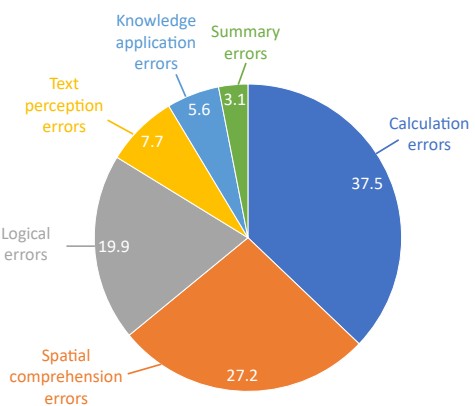

Figure 5: Error analysis for DouBao-1.5-Vision-Pro reveals six main issues: Calculation errors, Spatial comprehension errors, Logical errors, Text perception errors, Knowledge application errors, and Summary errors.

## 5 CONCLUSION

In this paper, we introduce OCR-Reasoning, a comprehensive benchmark designed to systematically evaluate the reasoning capabilities of state-of-the-art Multimodal Large Language Models (MLLMs) in text-rich image scenarios. The benchmark provides a structured framework comprising 1,069 human-annotated examples, which cover 6 core reasoning abilities across 18 distinct tasks. In contrast to existing benchmarks that focus solely on final-answer accuracy, OCR-Reasoning incorporates annotations for both the final answers and the corresponding step-by-step reasoning processes. This dual annotation enables a more comprehensive evaluation of model reasoning. Through extensive experiments, we reveal critical limitations of current MLLMs and identify potential avenues for future improvement.

**Limitations.** This work has two main limitations. First, since most of our data collection and annotation processes are performed manually, the high costs associated with these processes have resulted in our dataset size being comparable to previous methods (Liu et al., 2024d; Xu et al., 2025). In the future, we plan to combine automated annotation with human efforts to expand the dataset scale. Second, following the evaluation process of LLMs, we employed the LLMs-as-Judges approach to assess model reasoning processes. However, issues such as biases in LLMs-as-Judges, adversarial attacks, and inherent weaknesses in the methodology may affect evaluation accuracy (Li et al., 2024b). We intend to develop more robust evaluation approaches in future work.

ACKNOWLEDGMENTS

This research is supported in part by the National Natural Science Foundation of China (Grant No.:62476093).

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

## A APPENDIX

### A.1 THE USE OF LARGE LANGUAGE MODELS(LLMS)

We used LLMs only to polish and correct the grammar of this paper. This involved rephrasing sentences, checking grammar, and improving the overall flow for better readability and clarity. All research ideation, experiments, and analyses were developed by the authors, with no involvement from the LLM in these stages.

### A.2 QUALITATIVE ANALYSIS OF REASONING PATH

Here, we illustrate this with an example. As shown in Fig. 8(a), (b), and(c), we observe that in models such as Gemini and Claude-3.7-Sonnet, many errors occur where minor mistakes in the reasoning process lead to incorrect final answers, while the overall reasoning process remains fundamentally sound. Consequently, the LLM assigns a relatively high score to their problem-solving procedures.

### A.3 QUALITATIVE ANALYSIS OF OCR INPUT

We have provided some examples where OCR errors led to incorrect final results, as shown in Fig. 8(d). We found that LLMs generally struggle to provide correct answers when OCR results contain errors.

### A.4 QUALITATIVE ANALYSIS OF ERROR CASE

Fig. 6 presents a systematic comparison of MLLM reasoning processes. For OpenAI-o1, the main error lies in its failure to correctly understand the problem. In the scenario, the Green Salad had already been purchased and paid for, but OpenAI-o1 still included it when calculating the new price. As for DouBao-1.5-Vision-Pro, it exhibits similar issues to O1 and additionally made a mistake in understanding the product name. According to the menu, "6 jumbo shrimp" refers to a single dish instead of six individual jumbo shrimp. However, both Qwen2.5-VL-72B and DouBao-1.5-Vision-Pro misinterpret it as six separate food items. made the same error as DouBao-1.5-Vision-Pro.

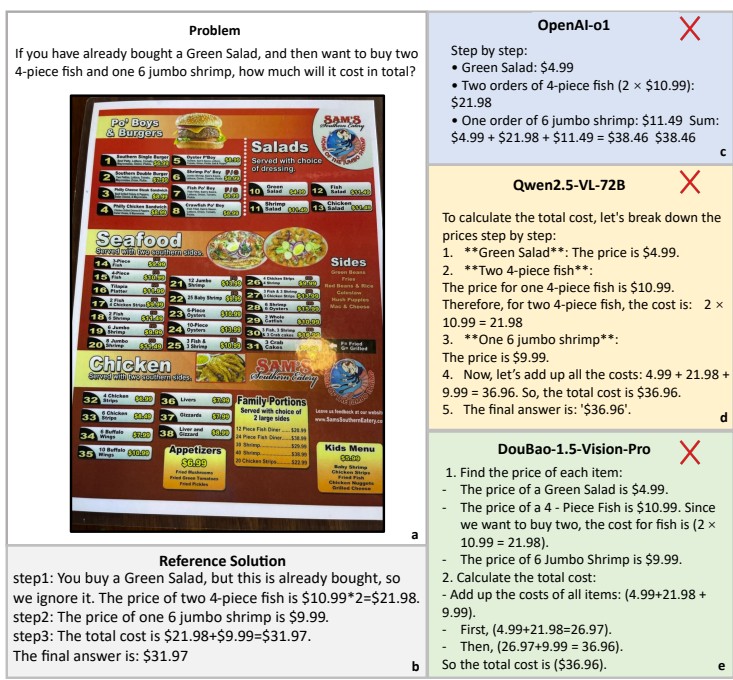

Figure 6: Solution examples generated by different models. (a) presents the input image and corresponding question; (b) shows the reference solution; (c)-(e) demonstrate outputs generated by different AI systems: (c) OpenAI-o1 model, (d) Qwen2.5-VL-72B model, and (e) DouBao-1.5-Vision-Pro model.

Table 7: Performance comparison of Qwen2.5-VL-7B before and after fine-tuning on the synthetic data generated by our data curation framework.

| Method | Overall | Spatial | Numerical Analysis | Mathematical | Enumerative | Logical | Multidisciplinary Knowledge |
|---|---|---|---|---|---|---|---|
| Qwen2.5-VL-7B (base model) | 15.7 | 13.8 | 11.6 | 8.8 | 20.1 | 9.0 | 35.8 |
| Qwen2.5-VL-7B (Fine-tune) | 17.0 | 14.7 | 10.6 | 21.6 | 22.9 | 17.4 | 26.3 |

## A.5 USE THE DATA CURATION FRAMEWORK TO ANNOTATE TRAINING DATA

We leveraged our annotation logic to replace human annotators with DouBao-1.5-Vision-Pro, selecting 5000 images from DocVQA, InfoVQA, TextVQA, and ChartQA for synthetic data generation. As presented in Tab. 7, the results of fine-tuning Qwen2.5-VL-7B using this synthetic data. These results demonstrate that the proposed data curation framework cannot only be used for building benchmark data but also be migrated to the construction of training datasets.

## A.6 A ROADMAP FOR OCR-REASONING

We outline a concrete roadmap for how OCR-Reasoning might evolve into a standardized benchmark suite for multimodal reasoning. We outline three core steps:

1. For reward design, we will leverage our annotation logic, replace the manual parts with leading models, and construct a batch of training data through standardized processes. Using this data, we will train a process-aware reward model that evaluates not only answer correctness but also the rationality of reasoning steps—delivering more granular, objective signals than rule-based reward functions. The model will be fully open-sourced to help the community systematically enhance models' reasoning capabilities on text-rich images.

2. We will make the benchmark open-source on Hugging Face to facilitate downloading and use by researchers in the community. Subsequently, we will open-source the code on commonly used

evaluation frameworks for MLLMs, enabling researchers to quickly evaluate their models' performance.

3. Existing benchmarks often focus on evaluating capabilities in a single dimension. This benchmark (OCR-Reasoning) focuses on assessing models' reasoning abilities on single images. Therefore, for dataset expansion, we will include more complex tasks such as reasoning on multilingual text, domain-specific documents, or multi-page handwritten materials.

### A.7 THE UNDERLYING MECHANISMS OF VL-RETHINKER-7B

When testing RL-trained models, we used their official prompts to ensure consistency between training and testing. On this basis, we further added the CoT prompt: 'Solve the complex problem through step-by-step reasoning.' We found that this approach improved performance on MM-Eureka, but degraded performance on VL-Rethinker. We hypothesize that this is due to a conflict between the generic CoT prompt and the model's built-in reflection/reasoning mechanism.

To further validate this, we conducted an ablation study by removing the model's official built-in prompt and then testing the effect of the generic CoT prompt. The results are presented below. The performance improved when the official prompt was removed (comparing the third and fourth rows). It supports our hypothesis that the degradation occurs specifically because the two prompts conflict within the model's input processing.

Table 8: Ablation study of the underlying mechanisms of VL-Rethinker-7B.

| Method | Official Prompt | CoT | Overall | Spatial | Numerical Analysis | Mathematical | Enumerative | Logical | Multidisciplinary Knowledge |
|---|---|---|---|---|---|---|---|---|---|
| Official Setting | ✓ | × | 19.1 | 13.7 | 16.6 | 9.8 | 25.7 | 14.6 | 33.6 |
| Original + CoT | ✓ | ✓ | 14.6 | 8.3 | 16.1 | 9.8 | 19.6 | 8.3 | 19.0 |
| Ablation 1 (No Prompt) | × | × | 17.5 | 13.8 | 15.8 | 7.8 | 21.2 | 16.7 | 28.5 |
| Ablation 2 (CoT) | × | ✓ | 18.7 | 14.7 | 13.1 | 11.7 | 27.9 | 18.8 | 31.3 |

### A.8 PRACTICAL REASONING TASKS

In this Section, we provide some examples of the 18 practical reasoning tasks in text-rich visual scenarios, as presented in Fig. 7.

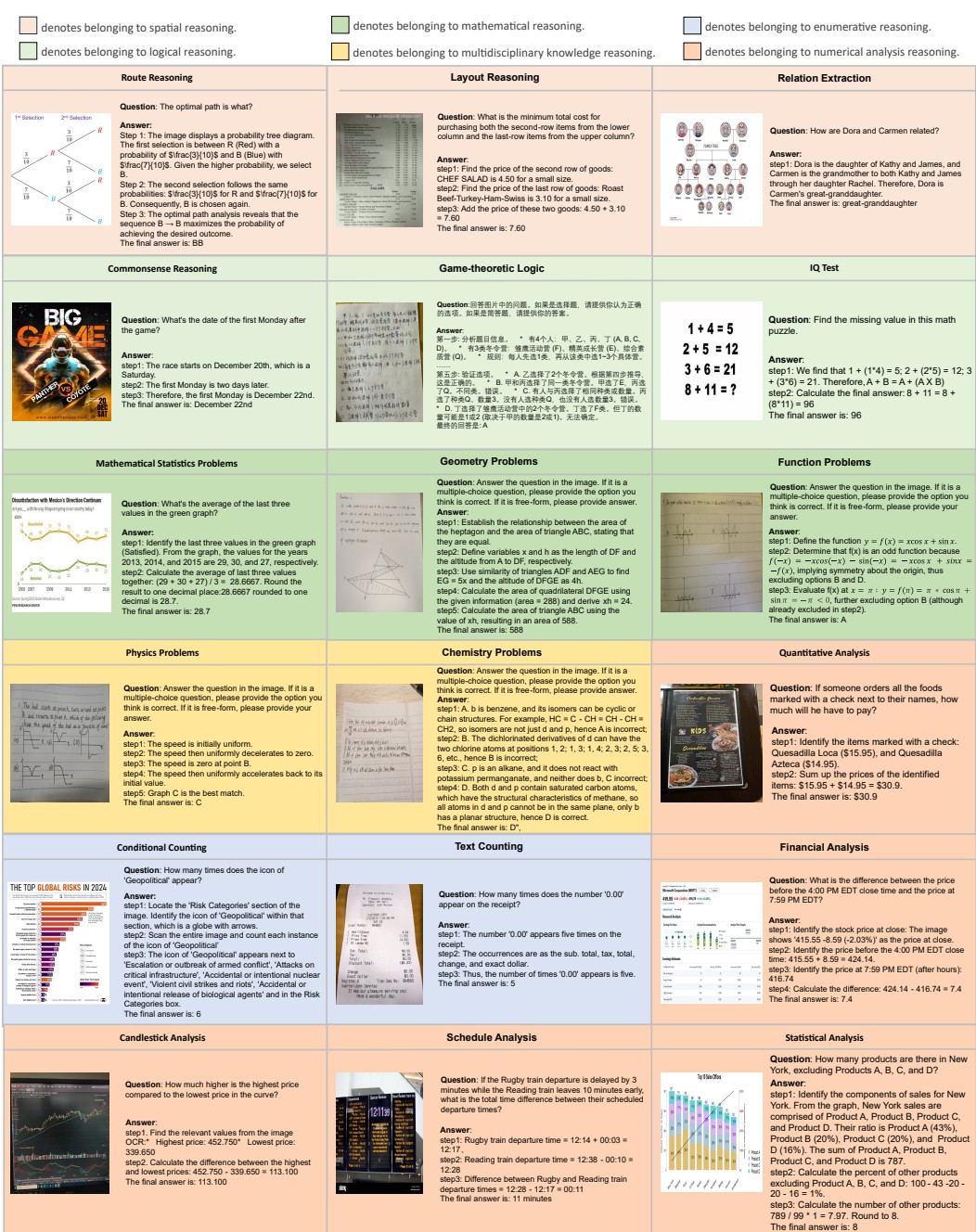

Figure 7: Examples of the 18 practical reasoning tasks in text-rich visual scenarios.

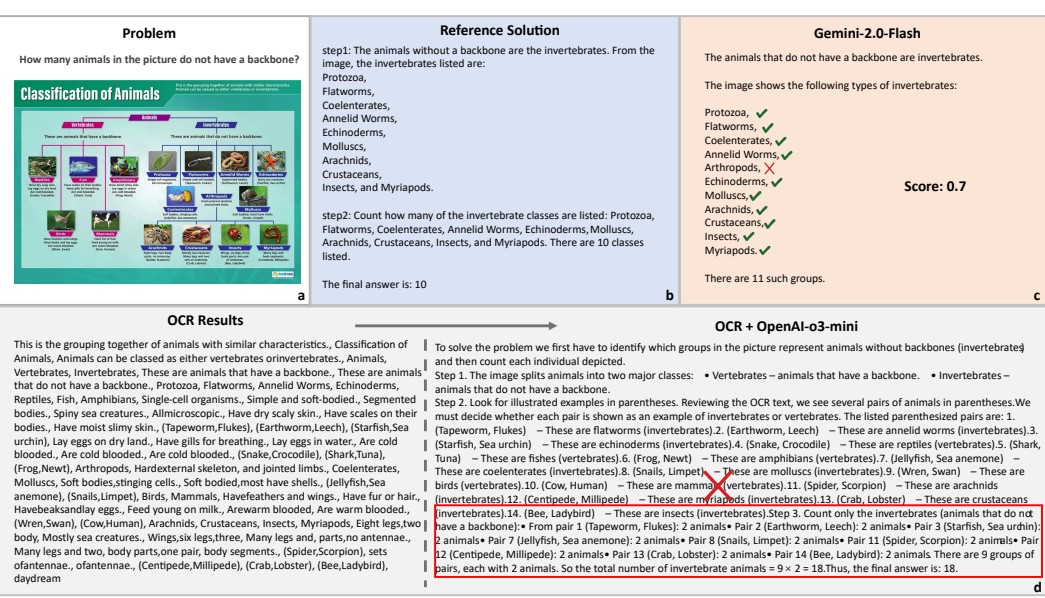

Figure 8: Qualitative analysis of the reasoning path. (a) presents the input image and corresponding question; (b) shows the reference solution serving as ground truth; (c) demonstrates outputs generated by Gemini-2.0-Flash. (d) demonstrates outputs generated by O3-mini with OCR results input.

