# OpenReview forum: "OCR-Reasoning Benchmark: Unveiling the True Capabilities of MLLMs in Complex Text-Rich Image Reasoning"
_ICLR.cc/2026/Conference — ICLR 2026 Poster_

### Official Review · Reviewer_KL6Q · 2025-10-26

**Soundness:** 2
**Presentation:** 2
**Contribution:** 2
**Rating:** 4
**Confidence:** 4

**Summary:**

This paper reveals severe capability gaps in current advanced multimodal slow-thinking systems when performing text-rich image reasoning tasks, attributing this limitation fundamentally to the absence of a systematic and specialized evaluation benchmark. To address this challenge, this paper introduces the OCR-Reasoning benchmark, characterized by its large-scale manual annotations, coverage of multi-dimensional reasoning tasks, and core feature of "dual annotation" providing both answers and detailed step-by-step reasoning processes.

**Strengths:**

This paper accurately identifies a crucial gap in multimodal reasoning research by proposing the first benchmark specifically designed for systematically evaluating "text-rich image reasoning" capabilities
﻿
The constructed OCR-Reasoning benchmark surpasses traditional models that focus solely on answer correctness. By incorporating annotations of step-by-step reasoning processes, it achieves dual evaluation of both model reasoning paths and final answers, providing a more comprehensive and profound perspective for diagnosing model capability shortcomings.

**Weaknesses:**

1. The study primarily focuses on diagnostic evaluation but does not validate the effectiveness of its proposed "thinking with images" reasoning approach. A critical question remains unanswered: can explicitly training models to generate such visual reasoning chains on the proposed benchmark lead to significant gains in reasoning performance? The paper would be significantly strengthened by including fine-tuning experiments that demonstrate whether and how leveraging this benchmark for training, as opposed to merely for evaluation, improves model capabilities on text-rich reasoning tasks.

2. The paper would benefit from a more thorough comparative analysis between the proposed OCR-Reasoning dataset and existing text-rich benchmarks (e.g., TextVQA, DocVQA, ChartQA). While the novel "dual annotation" is highlighted, a quantitative and qualitative comparison is needed to clearly delineate its advantages. This should explicitly detail the dataset's superiorities in terms of data quality (e.g., the depth and consistency of reasoning chain annotations), data characteristics (e.g., the diversity and complexity of reasoning types beyond simple QA), and coverage (e.g., the inclusion of scenarios that require multi-hop reasoning).

3. Although the benchmark's quality is commendable, its scale (approximately 1,069 examples) may limit its comprehensiveness and statistical power. The covered visual domains might not fully represent the vast spectrum of real-world text-rich images. To enhance the robustness and generalizability of the findings, the authors could consider expanding the dataset to include more diverse sources, such as web screenshots (for UI/UX reasoning), financial documents (for complex table and report understanding), academic papers, or product manuals. This would ensure that the benchmark tests reasoning capabilities across a broader range of practical contexts.

4. The benchmark is constructed manually, which ensures high quality but is not scalable for generating large-scale training data. A discussion on potential methodologies for scalable training data construction is a crucial missing piece. The authors could propose and discuss semi-automatic or synthetic data generation techniques that could produce large volumes of text-rich image reasoning data. For instance, exploring how advanced models (like GPT-4V) could assist in drafting reasoning chains for human review, or how to create synthetic tasks that inherently require visual-textual reasoning, would greatly enhance the practical utility and impact of this work beyond evaluation.

**Questions:**

see the weakness

---

> ### Author Response · Authors · 2025-11-23
> **Response to Reviewer KL6Q [1/3]**
>
> Thank you for your feedback on our work!
>
> ---
> ***Q1. The study primarily focuses on diagnostic evaluation but does not validate the effectiveness of its proposed "thinking with images" reasoning approach. A critical question remains unanswered: can explicitly training models to generate such visual reasoning chains on the proposed benchmark lead to significant gains in reasoning performance? The paper would be significantly strengthened by including fine-tuning experiments that demonstrate whether and how leveraging this benchmark for training, as opposed to merely for evaluation, improves model capabilities on text-rich reasoning tasks.***
>
> 1. Thanks for your comments. We would like to clarify that our study does not propose a "thinking with images" reasoning approach. Instead, its core contribution lies in introducing a challenging rich-text image reasoning benchmark, which cannot be used for training to obtain visual reasoning chains. Given that "thinking with images" represents a promising future direction, following your suggestion, we tested the performance of an open-source state-of-the-art "thinking with images" method on OCR-Reasoning. The experiment results are shown in the following Table. The experimental results demonstrate the potential of this approach for enhancing the model's text-rich image reasoning ability. We have supplemented these results in Section 4.2 of the revised manuscript.
>
> | Model                          | Overall | Spatial Reasoning | Numerical Reasoning | Mathematical Reasoning | Enumeration Reasoning | Logical Reasoning | Multidisciplinary Knowledge Reasoning |
> |--------------------------------|---------|-------------------|---------------------|------------------------|-----------------------|-------------------|---------------------------------------|
> | Qwen2.5-VL-7B (Base Model)     | 15.7    | 13.8              | 11.6                | 8.8                    | 20.1                  | 9.0                | 35.8                                  |
> | Thyme-RL-7B                    | 15.2    | 12.8              | 10.8                | 10.8                   | 20.7                  | 18.1               | 23.4                                  |
> | DeepEyesV2                     | 20.9    | 11.9              | 18.8                | 13.7                   | 27.9                  | 18.9               | 32.8                                  |
>
>
> 2. Additionally, we supplemented a fine-tuning experiment: leveraging our annotation logic, we replaced manual parts with DouBao-1.5-Vision-Pro and selected 5000 images from DocVQA, InfoVQA, TextVQA, and ChartQA to generate synthetic data. The results of fine-tuning Qwen2.5-VL-7B using this synthetic data are presented in the second line of the following table. These results demonstrate that the proposed data curation framework cannot only be used for building benchmark data but also be migrated to the construction of training datasets. These results have been added to Section A.5 of the appendix in the revised manuscript.
>
>
> | Model                          | Overall | Spatial Reasoning | Numerical Reasoning | Mathematical Reasoning | Enumeration Reasoning | Logical Reasoning | Multidisciplinary Knowledge Reasoning |
> |--------------------------------|---------|-------------------|---------------------|------------------------|-----------------------|-------------------|---------------------------------------|
> | Qwen2.5-VL-7B (Base Model)     | 15.7    | 13.8              | 11.6                | 8.8                    | 20.1                  | 9.0                | 35.8                                  |
> | Qwen2.5-VL-7B (Fine-tune)      | 17.0    | 14.7              | 10.6                | 21.6                   | 22.9                  | 17.4               | 26.3                                  |

---

> ### Author Response · Authors · 2025-11-23
> **Response to Reviewer KL6Q [2/3]**
>
> ---
> ***Q2. The paper would benefit from a more thorough comparative analysis between the proposed OCR-Reasoning dataset and existing text-rich benchmarks (e.g., TextVQA, DocVQA, ChartQA). While the novel "dual annotation" is highlighted, a quantitative and qualitative comparison is needed to clearly delineate its advantages. This should explicitly detail the dataset's superiorities in terms of data quality (e.g., the depth and consistency of reasoning chain annotations), data characteristics (e.g., the diversity and complexity of reasoning types beyond simple QA), and coverage (e.g., the inclusion of scenarios that require multi-hop reasoning).***
>
> Thanks for the suggestion. We supplement more thorough comparative analysis between the proposed OCR-Reasoning dataset and existing text-rich benchmarks.
>
> 1. Quantitative Comparison. We have added experimental results of comparison with existing datasets, based on Qwen2.5-VL-7B, InternVL3-8B, and TokenVL-8B. The experimental results are shown in the following table. We can see that even though existing methods perform well on existing datasets (e.g., DocVQA, ChartQA, TextVQA), they perform poorly on text-rich image reasoning tasks. The main reason is that existing datasets mainly evaluate the model's perceptual capabilities, while OCR-Reasoning requires the model to achieve accurate perception and further conduct thinking and reasoning. The text-rich image reasoning ability of existing models still has considerable room for improvement.
>
> | Models                | DocVQA | ChartQA | TextVQA | OCR-Reasoning |
> |-----------------------|-----------|-----------|-----------| -----------|
> | Qwen2.5-VL-7B  | 95.7   |  87.3  | 84.9   | 15.7   |
> | InternVL3-8B  | 92.7   | 86.6   | 80.2  | 11.5  |
> | TokenVL-8B   | 94.2  | 86.6     | 79.9  | 14.3  |
>
> 2. For data quality and data characteristics, our "dual annotation" (step-by-step reasoning chain + final answer) ensures depth and consistency. Each sample includes 3-6 reasoning steps verified by three annotators, while existing datasets only provide final answers without explicit reasoning processes. Furthermore, the dual annotation enables simultaneous evaluation of the reasoning process and the correctness of the final results, thereby facilitating a more comprehensive understanding of the model's capabilities. For coverage, our dataset covers six reasoning types with eighteen diverse scenarios that require multi-hop reasoning, far beyond the single-type text extraction of prior datasets.

---

> ### Author Response · Authors · 2025-11-23
> **Response to Reviewer KL6Q [3/3]**
>
> ---
> ***Q3. Although the benchmark's quality is commendable, its scale (approximately 1,069 examples) may limit its comprehensiveness and statistical power. The covered visual domains might not fully represent the vast spectrum of real-world text-rich images. To enhance the robustness and generalizability of the findings, the authors could consider expanding the dataset to include more diverse sources, such as web screenshots (for UI/UX reasoning), financial documents (for complex table and report understanding), academic papers, or product manuals. This would ensure that the benchmark tests reasoning capabilities across a broader range of practical contexts.***
>
>
> Thanks for your comments. OCR-Reasoning covers eighteen distinct real-world reasoning scenarios across six core reasoning abilities. This depth and diversity are comparable to other well-established, high-quality benchmarks in the field, such as OCRBench[1] (1,000 samples), the MathVista[2] testmini subset (1,000 samples), and VisuLogic[3] (1,000 samples). Additionally, this amount of data prioritizes depth and diversity, ensuring comprehensive coverage without excessive evaluation overhead. For web screenshots, financial documents, or product manuals mentioned by the reviewers, OCR-Reasoning has already included these scenarios and been classified under the Numerical Analysis Reasoning category. We have added relevant details in Section 3.1.
>
> ---
> ***Q4. The benchmark is constructed manually, which ensures high quality but is not scalable for generating large-scale training data. A discussion on potential methodologies for scalable training data construction is a crucial missing piece. The authors could propose and discuss semi-automatic or synthetic data generation techniques that could produce large volumes of text-rich image reasoning data. For instance, exploring how advanced models (like GPT-4V) could assist in drafting reasoning chains for human review, or how to create synthetic tasks that inherently require visual-textual reasoning, would greatly enhance the practical utility and impact of this work beyond evaluation.***
>
>
> Thanks for the discussion. As a benchmark, the quality of annotations is of great importance. To ensure this and the overall quality of the benchmark, we adopted extensive manual work. We did explore automatic annotation using advanced models. However, we found that for the complex, multi-step, text-rich reasoning tasks, current frontier models still perform poorly. This is also supported by experimental results: even the most advanced models face substantial difficulties in such tasks, with none exceeding 50% accuracy. While we cannot fully rely on advanced models for automatic annotation of benchmark, our data curation framework has already leveraged advanced model to assist in drafting reasoning chains for human review, thereby improving annotation efficiency and consistency, as detailed in Section 3.1 of the paper. Although current MLLM cannot directly achieve automatic annotation, with the advancement of MLLM and the continuous improvement of their capabilities in the future, it is highly probable that such a feat can be accomplished.
>
>
> For training data (where the tolerance for noise is higher than in a benchmark), the proposed data curation framework can be applied to synthetic data generation. We have also made some attempts, and the results are presented in Q1.

---

> > ### Comment · Reviewer_KL6Q · 2025-11-26
> > **resposne**
> >
> > Thanks for your response to my questions. I will raise my score from negative to positive.

---

> > > ### Author Response · Authors · 2025-11-26
> > >
> > > Thank you very much for your feedback and valuable suggestions. We sincerely appreciate your time and effort!

---

### Official Review · Reviewer_eZdh · 2025-10-26

**Soundness:** 3
**Presentation:** 3
**Contribution:** 4
**Rating:** 6
**Confidence:** 4

**Summary:**

The paper introduces the OCR-Reasoning benchmark, designed to evaluate the reasoning capabilities of Multimodal Large Language Models (MLLMs) in complex text-rich image scenarios. Unlike existing benchmarks that focus on text extraction, OCR-Reasoning includes both final answers and step-by-step reasoning processes for 1,069 annotated examples across six core reasoning categories. The benchmark reveals significant limitations in current MLLMs, with top models achieving only around 50% accuracy in reasoning tasks, highlighting struggles in integrating visual, textual, and logical information. The study emphasizes the importance of Chain-of-Thought prompting and provides valuable insights into common failure modes such as calculation, spatial comprehension, and logical errors. This benchmark aims to inspire future advancements in multimodal reasoning systems.

**Strengths:**

The paper demonstrates strong originality by introducing a new benchmark, OCR-Reasoning, that evaluates multimodal large language models on text-rich image reasoning—a domain previously underserved by existing datasets focused mainly on text extraction. Its dual annotation of final answers and reasoning steps represents a creative and meaningful extension of prior benchmarks.

In terms of quality, the study employs a rigorous and transparent methodology, including systematic dataset curation, expert annotation, and comprehensive evaluation across a broad spectrum of state-of-the-art models. The experimental design is sound, with clear baselines, detailed performance breakdowns, and thoughtful error analysis.

The paper exhibits strong clarity, with a well-structured presentation, clear motivation, and informative figures that effectively illustrate dataset design and results. The writing is precise and technically competent, enabling easy understanding of both the problem and its importance.

Regarding significance, the work makes a timely and impactful contribution to the multimodal reasoning community. By exposing the current limitations of MLLMs in integrating textual and visual reasoning, it establishes an essential benchmark that will likely guide future research on improving model reasoning and evaluation frameworks.

**Weaknesses:**

The dataset is relatively small (1,069 samples), limiting generalization and coverage of diverse real-world scenarios. The reliance on LLM-as-Judge introduces potential bias; incorporating human or cross-model validation would improve reliability. The paper lacks deeper diagnostic analysis explaining why models fail, and provides limited quantitative comparison with prior benchmarks. Finally, details on dataset release and reproducibility are insufficient, which may hinder adoption.

**Questions:**

1. Dataset Scale and Coverage:
Could the authors clarify whether there are plans to expand OCR-Reasoning beyond 1,069 samples? A larger and more diverse dataset (e.g., multilingual, domain-specific, or handwritten documents) would improve the benchmark’s representativeness.
2. Evaluation Bias in LLM-as-Judge:
How do the authors mitigate potential bias when using LLMs to evaluate reasoning quality, especially if the judging model shares architecture or training data with the tested models? Would cross-model or partial human evaluation be feasible for validation?
3. Reasoning-Type Analysis:
The paper shows category-wise results but lacks deeper diagnostics. Could the authors provide finer-grained error analyses or ablations—for example, separating failures due to OCR errors, visual reasoning, or logical inference?
4. Comparison with Existing Benchmarks:
It would be helpful to include a more direct experimental comparison or transfer evaluation with existing datasets (e.g., DocVQA, ChartQA, OCRBench). How does OCR-Reasoning specifically challenge models beyond these benchmarks?
5. Reinforcement Learning Methods:
The paper notes that RL-based methods perform poorly. Could the authors elaborate on how a better reward design for text-rich reasoning might look, or what specific factors caused these RL models to fail?
6. Dataset Accessibility and Reproducibility:
Please clarify the intended release details—license, format, annotation schema, and evaluation scripts. Ensuring full reproducibility will significantly strengthen the paper’s long-term impact.
7. Future Directions:
The authors mention possible improvements in reward design and dataset expansion. Could they outline a concrete roadmap for how OCR-Reasoning might evolve into a standardized benchmark suite for multimodal reasoning?
These clarifications and extensions could meaningfully strengthen the paper’s rigor, reproducibility, and long-term value to the community.

---

> ### Author Response · Authors · 2025-11-23
> **Response to Reviewer eZdh [1/3]**
>
> Thank you very much for your thoughtful feedback on our work!
>
> ---
> ***Q1. The dataset is relatively small (1,069 samples), limiting generalization and coverage of diverse real-world scenarios.***
>
>
> We acknowledge the concern regarding scale. However, OCR-Reasoning covers 18 distinct real-world reasoning scenarios across 6 core reasoning abilities. This depth and diversity are comparable to other well-established, high-quality benchmarks in the field, such as OCRBench[1] (1,000 samples), the MathVista[2] testmini subset (1,000 samples), and VisuLogic[3] (1,000 samples). Additionally, this amount of data prioritizes depth and diversity, ensuring comprehensive coverage without excessive evaluation overhead. Moreover. annotating the high-quality OCR-Reasoning that includes both reasoning processes and answers is challenging and highly labor-intensive. It involved the efforts of 10 professional annotators, and the entire annotation process took 4 months to reach final completion. We plan to further expand the dataset at an appropriate opportunity in the future. We sincerely appreciate the reviewer's valuable comment.
>
> [1] Liu Y, Li Z, Huang M, et al. OCRBench: On the Hidden Mystery of OCR in Large Multimodal Models[J]. Science China Information Sciences, 2024, 67(12): 220102.
>
> [2] Lu P, Bansal H, Xia T, et al. MathVista: Evaluating Mathematical Reasoning of Foundation Models in Visual Contexts[C]//The Twelfth International Conference on Learning Representations.
>
> [3] Xu W, Wang J, Wang W, et al. VisuLogic: A Benchmark for Evaluating Visual Reasoning in Multi-modal Large Language Models[J]. arXiv e-prints, 2025: arXiv: 2504.15279.
>
> ---
> ***Q2. Dataset Scale and Coverage: Could the authors clarify whether there are plans to expand OCR-Reasoning beyond 1,069 samples? A larger and more diverse dataset (e.g., multilingual, domain-specific, or handwritten documents) would improve the benchmark’s representativeness.***
>
> Thank you for the suggestion. We agree that future expansion is valuable. This current version of OCR-Reasoning is designed to evaluate single-image reasoning capabilities. We are actively planning an OCR-Reasoning v2 that will extend to more complex scenarios, including multilingual documents, domain-specific or multi-page handwritten documents.
>
>
> ---
> ***Q3. Evaluation Bias in LLM-as-Judge: How do the authors mitigate potential bias when using LLMs to evaluate reasoning quality, especially if the judging model shares architecture or training data with the tested models? Would cross-model or partial human evaluation be feasible for validation?***
>
> 1. We only employed the LLM-as-judge approach to assess the reasoning process. As shown in Section 3.3 of the paper, our adoption of LLM-as-Judge is based on solid empirical justification: 1). Human evaluation is costly, while LLM can quickly process large amounts of data. 2). LLM can reduce the variance among human evaluators. 3). LLM as Judge is commonly used in NLP to evaluate reasoning processes.
>
> 2. Additaionally, to address concerns about judge bias and enhance the generalizability and robustness of the assessment methodology, we have extended the human validation to three additional representative models: Qwen2.5-VL-72B, Llama4-Scout-109B-A17B, and OpenAI-o1. The results consistently demonstrate the alignment between human assessments and LLM-generated scores across different models.
>
> | Model | Human Validation Score | LLM-as-Judge Score |
> | :--- | :--- | :--- |
> | Qwen2.5-VL-72B | 50.2 | 51.8 |
> | Llama4-Scout-109B-A17B | 43.8 | 44.9 |
> | OpenAI-o1 | 47.6 | 48.5  |
> | DouBao-1.5-Vision-Pro | 53.1 | 55.4 |
>
> 3. Thanks for the suggestion. To address the concern about judge-model bias, we have conducted a cross-model validation for DouBao-1.5-Vision-Pro, using GPT-4o, Gemini-2.5-flash and Qwen3-max. The results are presented in the following table. These results show that while different judge models yield varying final results, the discrepancies between them are acceptable.
>
> | Judge Model | Overall | Spatial | Numerical Analysis | Mathematical | Enumerative | Logical | Multidisciplinary Knowledge |
> | --- | --- | --- | --- | --- | --- | --- | --- |
> | GPT-4o | 55.0 | 37.9 | 61.4 | 50.1 | 51.8 | 52.1 | 60.8 |
> | Gemini-2.5-flash | 56.2 | 38.8 | 62.3 | 50.9 | 53.2 | 53.7 | 62.5 |
> | Qwen3-max | 55.8 | 39.1 | 61.6 | 50.5 | 53.7 | 53.3 | 61.4 |

---

> > ### Author Response · Authors · 2025-11-23
> > **Response to Reviewer eZdh [2/3]**
> >
> > ---
> > ***Q4. Reasoning-Type Analysis: The paper shows category-wise results but lacks deeper diagnostics. Could the authors provide finer-grained error analyses or ablations—for example, separating failures due to OCR errors, visual reasoning, or logical inference?***
> >
> > Thanks for the comment. We have performed a fine-grained error analysis on a representative sample of model failures in Section 4.4 of the paper, categorize the primary causes of failure as follows: (1) calculation errors (37.5\%), (2) spatial comprehension errors (27.2\%), (3) logical errors (19.9\%), (4) text perception errors (7.7\%), (5) knowledge application errors (5.6\%), and (6) summary errors (3.1\%).
> >
> >
> > ---
> > ***Q5. Comparison with Existing Benchmarks: It would be helpful to include a more direct experimental comparison or transfer evaluation with existing datasets (e.g., DocVQA, ChartQA, OCRBench). How does OCR-Reasoning specifically challenge models beyond these benchmarks?***
> >
> > Thanks for the suggestion. We have supplemented experiments to compare OCR-Reasoning with existing datasets, using Qwen2.5-VL-7B, InternVL3-8B, and TokenVL-8B as base models. The experimental results are shown in the following table. We can see that even though existing methods perform well on existing datasets (e.g., DocVQA, ChartQA, OCRBench), they perform poorly on text-rich image reasoning tasks. The main reason may be that existing datasets mainly evaluate the model's perceptual capabilities, while OCR-Reasoning requires the model to achieve accurate perception and further conduct thinking and reasoning. The text-rich image reasoning ability of existing models still has considerable room for improvement.
> >
> > | Models                | DocVQA | ChartQA | OCRBench | OCR-Reasoning |
> > |-----------------------|-----------|-----------|-----------| -----------|
> > | Qwen2.5-VL-7B  | 95.7   |  87.3  | 864   | 15.7   |
> > | InternVL3-8B  | 92.7   | 86.6   | 880   | 11.5  |
> > | TokenVL-8B   | 94.2  | 86.6     | 860  | 14.3  |
> >
> > We add these results in Section 4.3 of the revised manuscript.
> >
> > ---
> > ***Q6. Reinforcement Learning Methods: The paper notes that RL-based methods perform poorly. Could the authors elaborate on how a better reward design for text-rich reasoning might look, or what specific factors caused these RL models to fail?***
> >
> > Thanks for the disscusion. As presented in Section 2, the poor performance of existing RL models may be stem from two factors:
> >
> > 1. Data domain mismatch. OCR-Reasoning covers more real-world text-rich scenarios. In contrast, the training data of existing RL methods usually consists of printed individual math problems, leading to a distribution gap between training data and downstream tasks.
> > 2. Most reward functions in exsting methods are tailored for mathematical reasoning tasks and lack designs for text-rich scenarios.
> >
> > For reward design optimization in text-rich reasoning, we suggest that the potential directions for improvement may be:
> >
> > 1. Multi-dimensional Composite Reward. Text-rich image reasoning tasks require models to accurately understand the layout information of text and the corresponding text content before conducting further reasoning. Therefore, we can design a reward function to enhance the model's acquisition of text information and answer accuracy simultaneously, thereby improving the model's reasoning ability.
> >
> > 2. Develop a process reward model. Leverage our annotation logic, replace the manual parts with leading models, and construct a batch of training data with processes. This dataset will train a PRM that evaluates not only the final answer but also the rationality of intermediate reasoning steps—enabling more fine-grained and accurate reward signals compared to traditional terminal rewards.

---

> > > ### Author Response · Authors · 2025-11-23
> > > **Response to Reviewer eZdh [3/3]**
> > >
> > > ---
> > > ***Q7. Dataset Accessibility and Reproducibility: Please clarify the intended release details—license, format, annotation schema, and evaluation scripts. Ensuring full reproducibility will significantly strengthen the paper’s long-term impact.***
> > >
> > > Thanks for the suggestion.
> > >
> > > 1. OCR-Reasoning is licensed under CC BY-NC-SA 4.0.
> > >
> > > 2. The format of OCR-Reasoning including image, question, answer, the format of answer, and the type of ability evaluated by this sample. The annotation schema is structured. Each sample contains the content mentioned above, and then all the annotated information is placed into a single JSON file.
> > >
> > > 3. As detailed in Section 3.3，the evaluation scripts is following mathvista. OCR-Reasoning adopts a threestage evaluation framework: (1) Response Generation, (2) Answer Extraction, and (3) Score Computation. First, the multimodal large language model (MLLM) processes an input query to generate detailed responses. Subsequently, an LLM-based answer extractor (e.g., GPT-4o) extracts concise answer text from these responses through semantic parsing. Finally, the extracted answers undergo normalization into standardized formats (e.g., option letters, integers, or strings) before accuracy-based metric calculation for deterministic evaluation.
> > >
> > > We will open-source our dataset and integrate the evaluation scripts to facilitate the community in evaluating their own models and ensure reproducibility.
> > >
> > >
> > > ---
> > > ***Q8. Future Directions: The authors mention possible improvements in reward design and dataset expansion. Could they outline a concrete roadmap for how OCR-Reasoning might evolve into a standardized benchmark suite for multimodal reasoning? These clarifications and extensions could meaningfully strengthen the paper’s rigor, reproducibility, and long-term value to the community.***
> > >
> > > Thanks for the suggestion. We outline three core steps:
> > >
> > > 1. For reward design, we will leverage our annotation logic, replace the manual parts with leading models, and construct a batch of training data with processes. Using this data, we will train a process reward model that evaluates not only answer correctness but also the rationality of reasoning steps—delivering more granular, objective signals than rule-based reward function. The model will be fully open-sourced to help the community systematically enhance models’ reasoning capabilities on text-rich images.
> > >
> > > 2. We will make the benchmark open-source to facilitate downloading and use by researchers in the community. Then, we will open source the code on commonly used evaluation frameworks for MLLMs, so that researchers can quickly evaluate their models' performance.
> > >
> > > 3. One benchmark focuses on evaluting capabilities in one dimension. This dataset focuses on evaluating models' reasoning ability on single images. Therefore, for dataset expansion, we will includes more complex tasks such as reasoning on multilingual, domain-specific or multi-page handwritten documents.
> > >
> > > These disscusion have been incorporated into Section A.6 of the revised manuscript.

---

### Official Review · Reviewer_YFNZ · 2025-10-27

**Soundness:** 3
**Presentation:** 3
**Contribution:** 3
**Rating:** 8
**Confidence:** 4

**Summary:**

This paper proposes OCR-Reasoning, a benchmark for evaluating MLLMs’ text-rich image reasoning, with 1,069 human-annotated examples (6 core reasoning abilities, 18 tasks) and annotations of both reasoning processes and answers (unlike existing benchmarks only annotating answers). Evaluations show top closed-source MLLM  doesn’t exceed 50% on it, while open-source ones perform worse.

**Strengths:**

1. Filling text-rich image reasoning evaluation gaps: Existing text-rich image benchmarks focus on text extraction but lack systematic reasoning assessment. OCR-Reasoning addresses this, measuring MLLMs’ reasoning in practical scenarios.
2. Sample design forcing reasoning: Few answers in its samples are directly extractable from OCR results; models must actively reason, avoiding reliance on text extraction to truly reflect their reasoning levels.
3. Comprehensive annotations for in-depth evaluation: Unlike benchmarks that only annotate final answers, OCR-Reasoning  labels both reasoning processes and answers, enabling holistic analysis of models’ problem-solving abilities

**Weaknesses:**

1. Limited dataset scale: Most of the data collection and annotation processes rely on manual work, and the high associated costs result in the dataset scale being only comparable to previous methods, failing to achieve larger-scale expansion

**Questions:**

1. OCR-Reasoning annotates both reasoning processes and final answers, while existing text-rich image benchmarks (e.g., DocVQA, OCRBench) mostly only annotate final answers and their samples’ answers are often directly extractable from OCR results. What are the specific core differences between OCR-Reasoning and these benchmarks in terms of sample design and annotation logic? What key role does this difference play in evaluating the true reasoning capabilities of MLLMs?
2. The paper states that existing reinforcement learning (RL) methods perform poorly on OCR-Reasoning, due to mismatched reward functions and a disconnect between training data and the benchmark’s scenarios. Does the study propose preliminary improvement directions (e.g., specific reward function design ideas, training data selection criteria)?

---

> ### Author Response · Authors · 2025-11-23
> **Response to Reviewer YFNZ [1/1]**
>
> Thank you very much for your thoughtful feedback on our work!
>
> ---
> ***Q1. Limited dataset scale: Most of the data collection and annotation processes rely on manual work, and the high associated costs result in the dataset scale being only comparable to previous methods, failing to achieve larger-scale expansion.***
>
> We appreciate this observation. However, as a high-quality benchmark, the fidelity of our annotations is paramount. To ensure this, we prioritized extensive manual curation. We did explore semi-automated annotation using advanced MLLMs for data scaling, but found that these models performed poorly on generating valid reasoning chains for our tasks, demonstrating that they are unsuitable for our benchmark construction. This is also supported by experimental results: even the most advanced models face substantial difficulties in such tasks, with none exceeding 50% accuracy on our benchmark.
>
>
> ---
> ***Q2. OCR-Reasoning annotates both reasoning processes and final answers, while existing text-rich image benchmarks (e.g., DocVQA, OCRBench) mostly only annotate final answers and their samples’ answers are often directly extractable from OCR results. What are the specific core differences between OCR-Reasoning and these benchmarks in terms of sample design and annotation logic? What key role does this difference play in evaluating the true reasoning capabilities of MLLMs?***
>
> Thanks for this important question. The core differences are as follows::
>
> 1. Sample design: Existing benchmarks do not require special considerations for image sample selection, as they primarily focus on perception-oriented capabilities. In contrast, when collecting images for OCR-Reasoning, we deliberately select images rich in textual and layout details that can support the design of non-trivial, reasoning-intensive questions.
>
> 2. Annotation logic: The questions in existing benchmarks are primarily designed for information extraction tasks where answers are often directly extractable from OCR results, and they typically only annotate the final answers. In contrast, the questions in our OCR-Reasoning benchmark are meticulously designed to require multiple reasoning steps (e.g., calculation, comparison, synthesis) before the final answer can be derived. Therefore, we annotate both reasoning processes and final answers to facilitate a more in-depth evaluation of MLLMs’ reasoning capabilities.
>
> These fundamental differences shift the evaluation focus from mere perception to holistic reasoning, enabling a deeper analysis of an MLLM's problem-solving fidelity and its ability to integrate visual, textual, and logical information for OCR reasoning.
>
> ---
> ***Q3. The paper states that existing reinforcement learning (RL) methods perform poorly on OCR-Reasoning, due to mismatched reward functions and a disconnect between training data and the benchmark’s scenarios. Does the study propose preliminary improvement directions (e.g., specific reward function design ideas, training data selection criteria)?***
>
> Thanks for the disscussion! We propose the following preliminary directions for improvement:
>
> For Data Selection:
>
> 1. A preliminary direction is to implement a Reasoning-Filter for existing data. This filter would select training examples where the answer is not directly present within the image's raw OCR results, thereby forcing the model to learn genuine reasoning capabilities rather than simple information extraction.
>
> For Reward Design:
>
> 1. Develop a Process Reward Model (PRM). By leveraging our dual-annotation logic (reasoning steps + answer), we can use new state-of-the-art models to semi-automatically construct a large-scale dataset. This data can be then used to train a dedicated Process Reward Model (PRM), enabling the calculation of more granular and accurate rewards based on the intermediate steps of the model's reasoning.
>
> 2. Design a Multi-dimensional Composite Reward. For complex text-rich tasks, the model should succeed at both low-level (OCR/Layout perception) and high-level (Reasoning) tasks. A composite reward function could be designed to simultaneously reward the model for (1) correctly identifying and localizing relevant textual information and (2) producing the correct final answer.

---

### Official Review · Reviewer_cM84 · 2025-11-03

**Soundness:** 4
**Presentation:** 4
**Contribution:** 4
**Rating:** 8
**Confidence:** 4

**Summary:**

This work addresses the lack of systematic benchmarks for evaluating Multimodal Large Language Models (MLLMs) in text-rich image reasoning. It proposes OCR-Reasoning, a benchmark with 1,069 human-annotated examples covering 6 core reasoning abilities and 18 practical tasks, featuring both final answers and step-by-step reasoning processes. Extensive evaluations of LLMs, MLLMs, and document-oriented MLLMs show that no model achieves over 50% accuracy, with image input outperforming OCR text alone and CoT prompting benefiting most models. The core contributions include the first reasoning process-annotated benchmark for text-rich images, systematic model evaluation, and identification of key improvement directions.

**Strengths:**

1. OCR-Reasoning is the first benchmark to systematically assess reasoning processes in text-rich image scenarios, addressing a long-overlooked need.
2. The comprehensive evaluation includes multiple model categories and zero-shot settings, ensuring generalizable results.
3. Detailed error analysis and qualitative case studies deepen understanding of model limitations beyond accuracy metrics.

**Weaknesses:**

1. While the handwritten data in OCR-Reasoning provides valuable transcribed college-level STEM problems, it would be beneficial to consider incorporating more everyday real-world handwritten scenarios to further enhance the benchmark's coverage of diverse text-rich reasoning tasks commonly encountered in practice.

2. The paper presents an interesting observation that CoT prompting may have backfired on VL-Rethinker-7B, potentially due to conflicting built-in reflection mechanisms. It would strengthen this finding if the authors could provide additional ablation studies or experiments to further validate this hypothesis and better understand the underlying mechanisms.

3. The human validation for the LLM-as-Judge method demonstrates careful evaluation on DouBao-1.5-Vision-Pro. To further establish the robustness of this evaluation approach, it would be valuable to extend the validation across additional models and reasoning categories, which could help address potential concerns about judge bias and generalizability of the assessment methodology.

**Questions:**

The zero-shot evaluation effectively demonstrates out-of-the-box model capabilities. Have the authors explored or considered exploring few-shot prompting or fine-tuning scenarios on OCR-Reasoning? Could such experiments provide insights into whether models achieve substantial improvements with modest amounts of task-specific guidance?

---

> ### Author Response · Authors · 2025-11-23
> **Response to Reviewer cM84 [1/2]**
>
> Thank you very much for your thoughtful feedback on our work!
>
> ---
> ***Q1. While the handwritten data in OCR-Reasoning provides valuable transcribed college-level STEM problems, it would be beneficial to consider incorporating more everyday real-world handwritten scenarios to further enhance the benchmark's coverage of diverse text-rich reasoning tasks commonly encountered in practice.***
>
> Thanks for the suggestion. We agree that incorporating everyday handwritten scenarios from real-world contexts is crucial. In addition to expert college-level STEM problems, OCR-Reasoning also includes a number of handwritten data about real-world logical reasoning tasks. We have now clarified and detailed these additions in Section 3.1 of the revised manuscript.
>
> ---
> ***Q2. The paper presents an interesting observation that CoT prompting may have backfired on VL-Rethinker-7B, potentially due to conflicting built-in reflection mechanisms. It would strengthen this finding if the authors could provide additional ablation studies or experiments to further validate this hypothesis and better understand the underlying mechanisms.***
>
> Thanks for this comment. In our initial evaluation of RL-trained models, we used their official system prompts to ensure consistency between training and testing. On this basis, we further added the CoT prompt: 'Solve the complex problem through step-by-step reasoning.' We found that this approach improved performance on MM-Eureka, but degraded performance on VL-Rethinker. We hypothesize that this is due to a conflict between the generic CoT prompt and the model's built-in reflection/reasoning mechanism.
>
> To further validate this, we conducted an ablation study by removing the model's official built-in prompt and then testing the effect of the generic CoT prompt. The results are presented below. The performance improved when the official prompt was removed (comparing the third and fourth rows). It supports our hypothesis that the degradation occurs specifically because the two prompts conflict within the model's input processing. We have added relevant details in Section A.7 of the revised manuscript.
>
> | Method               | Official Prompt | CoT  | Overall | Spatial | Numerical Analysis | Mathematical | Enumerative | Logical | Multidisciplinary Knowledge |
> | -------------------- | ---------------- | ---- | ------- | ------- | ------------------ | ------------ | ----------- | ------- | --------------------------- |
> | Official Setting     | ✓                | ×    | 19.1    | 13.7    | 16.6               | 9.8          | 25.7        | 14.6    | 33.6                        |
> | Original + CoT       | ✓                | ✓    | 14.6    | 8.3     | 16.1               | 9.8          | 19.6        | 8.3     | 19.0                        |
> | Ablation 1 (No Prompt)    | ×            | ×    | 17.5    | 13.8    | 15.8               | 7.8          | 21.2        | 16.7    | 28.5                        |
> | Ablation 2 (CoT) | ×            | ✓    | 18.7    | 14.7    | 13.1               | 11.7         | 27.9        | 18.8    | 31.3                        |
>
>
>
> ---
> ***Q3. The human validation for the LLM-as-Judge method demonstrates careful evaluation on DouBao-1.5-Vision-Pro. To further establish the robustness of this evaluation approach, it would be valuable to extend the validation across additional models and reasoning categories, which could help address potential concerns about judge bias and generalizability of the assessment methodology.***
>
> Thank you for this insightful suggestion. To address concerns about judge bias and enhance the generalizability and robustness of the assessment method, we have extended the human validation to three additional representative models: Qwen2.5-VL-72B, Llama4-Scout-109B-A17B, and OpenAI-o1. The results shown below consistently demonstrate the alignment between human assessments and LLM-generated scores across different models. These findings have been incorporated into Section 3.3 of the revised manuscript.
>
> | Model | Human Validation Score | LLM-as-Judge Score |
> | :--- | :--- | :--- |
> | Qwen2.5-VL-72B | 50.2 | 51.8 |
> | Llama4-Scout-109B-A17B | 43.8 | 44.9 |
> | OpenAI-o1 | 47.6 | 48.5  |
> | DouBao-1.5-Vision-Pro | 53.1 | 55.4 |

---

> > ### Author Response · Authors · 2025-11-23
> > **Response to Reviewer cM84 [2/2]**
> >
> > ---
> > ***Q4. The zero-shot evaluation effectively demonstrates out-of-the-box model capabilities. Have the authors explored or considered exploring few-shot prompting or fine-tuning scenarios on OCR-Reasoning? Could such experiments provide insights into whether models achieve substantial improvements with modest amounts of task-specific guidance?***
> >
> > Thanks for the suggestion. We supplemented experiments on few-shot prompting. Specifically, we annotated three additional samples as few-shot demonstrations for each reasoning capability and validated the performance of one-shot and three-shot prompting on Qwen2.5-VL-7B. As shown in the table below, few-shot prompting improves overall performance—particularly on subtasks requiring adherence to specific logical steps (e.g., Numerical Analysis Reasoning and Logical Reasoning). This demonstrates that moderate task-specific guidance helps the model understand and comply with task requirements. We observed a decline in the performance of Multidisciplinary Knowledge Reasoning. The potential reasons may be: The increased length of input tokens caused by few-shot examples, combined with the extended reasoning path inherently required by multidisciplinary knowledge reasoning, poses a significant challenge to the model's long-text processing and reasoning capabilities. These results have been added to Section 4.2 of the revised manuscript.
> >
> > | Few-Shot Prompting  | Overall | spatial reasoning | numerical analysis reasoning | mathematical reasoning | enumerative reasoning | logical reasoning | multidisciplinary knowledge reasoning |
> > | ---------------------------------- | ------- | ----------------- | ---------------------------- | ---------------------- | --------------------- | ------------------ | -------------------------------------- |
> > | Qwen2.5-VL-7B (base model) | 15.7  | 13.8  | 11.6  | 8.8  | 20.1  | 9.0  | 35.8 |
> > | One-shot       | 16.1 | 12.8 | 14.6 | 10.8 | 22.3 | 13.9 | 21.1   |
> > | Three-shot     | 16.4 | 13.7 | 14.8   | 10.0 | 22.9 | 13.2 | 22.3   |

---

### Meta-Review · Area_Chair_Tn6C · 2026-01-04

**Summary:**

This paper introduces OCR-Reasoning, a text-rich image reasoning benchmark intended to probe the true reasoning abilities of modern MLLMs beyond straightforward OCR or information extraction. One strong aspect is that the benchmark includes dual annotations: both final answers and step-by-step reasoning traces, enabling evaluation of outcome correctness and process quality.

Overall, AC agrees that the paper is a solid benchmark contribution that fills a practical evaluation gap for text-rich image reasoning and is likely to be used by the community, and the rebuttal meaningfully strengthened the empirical story and addressed major reviewer concerns. However, due to the primarily incremental nature of the contribution (benchmark + evaluation, limited new methodology) and remaining constraints on scale/coverage, AC does not see it as a strong candidate for higher-impact oral/spotlight. A poster acceptance is appropriate.

**Reviewer Concerns:**

First, dataset scale and coverage remain a reasonable limitation. The authors gave a reasonable explanation for why the benchmark is small and showed that its size is comparable to other curated datasets, but this does not fully address the concern. With about one thousand examples, the benchmark is strong as a diagnostic tool, but it cannot fully represent the diversity of real-world text-rich images. AC think the claims about general or “true” capabilities of MLLMs need to stay modest in the final draft.

Next, the contribution is mainly evaluative rather than methodological. The rebuttal added some fine-tuning and “thinking with images” experiments, which helped, but the gains were limited and preliminary. In practice, the benchmark is most convincing as a way to reveal failure modes, not as a demonstrated path to significantly improving models IMO. This is okay, but it caps the paper’s impact, so at most a poster.

Last, scalability and long-term impact are open questions. The authors explained why manual annotation was necessary and why current models are not reliable enough for full automation, AC feel it makes sense. However, this also means it is unclear how easily the benchmark can grow or become a long-term standard without significant additional effort. This does not undermine the current contribution, but it does limit how far the work can go.

**Reviewer Scores:**

KL6Q will likely raise score to 6.

Others remain the same.

So final ratings, 8 8 6 6

---

### Decision · Program_Chairs · 2026-01-26

Accept (Poster)